# Range, not Independence, Drives Modularity in Biologically Inspired Representations

**Will Dorrell**[*,1]     **Kyle Hsu**[*,2]     **Luke Hollingsworth**[†,1]     **Jin Hwa Lee**[†,1]     **Jiajun Wu**[2]

**Chelsea Finn**[2]     **Peter E. Latham**[1]     **Tim Behrens**[1,3]     **James C.R. Whittington**[2,3]

[1]University College London     [2]Stanford University     [3]Oxford University     [*]Co-First     [†]Co-Second

## Abstract

Why do biological and artificial neurons sometimes modularise, each encoding a single meaningful variable, and sometimes entangle their representation of many variables? In this work, we develop a theory of when biologically inspired networks—those that are nonnegative and energy efficient—modularise their representation of source variables (sources). We derive necessary and sufficient conditions on a sample of sources that determine whether the neurons in an optimal biologically-inspired linear autoencoder modularise. Our theory applies to any dataset, extending far beyond the case of statistical independence studied in previous work. Rather we show that sources modularise if their support is "sufficiently spread". From this theory, we extract and validate predictions in a variety of empirical studies on how data distribution affects modularisation in nonlinear feedforward and recurrent neural networks trained on supervised and unsupervised tasks. Furthermore, we apply these ideas to neuroscience data, showing that range independence can be used to understand the mixing or modularising of spatial and reward information in entorhinal recordings in seemingly conflicting experiments. Further, we use these results to suggest alternate origins of mixed-selectivity, beyond the predominant theory of flexible nonlinear classification. In sum, our theory prescribes precise conditions on when neural activities modularise, providing tools for inducing and elucidating modular representations in brains and machines.

## 1 Introduction

Our brains are modular. At the macroscale, different regions, such as visual or language cortex, perform specialised roles; at the microscale, single neurons often precisely encode single variables such as self-position (Hafting et al., 2005) or the orientation of a visual edge (Hubel and Wiesel, 1962). This mysterious alignment of meaningful concepts with single neuron activity has for decades fuelled hope for understanding a neuron's function by finding its associated concept. Yet, as neural recording technology has improved, it has become clear that many neurons behave in ways that elude such simple categorisation: they appear to be mixed selective, responding to a mixture of variables in linear and nonlinear ways (Rigotti et al., 2013; Tye et al., 2024). The modules vs. mixtures debate has recently been reprised in the machine learning community. Both the disentangled representation learning and mechanistic interpretability subfields are interested in when neural network representations decompose into meaningful components. Findings have been similarly varied, with some studies showing meaningful single unit response properties and others clearly showing mixed tuning (for a full discussion on related work see Appendix A). This brings us to our main research question: Why are neurons, biological and artificial, sometimes modular and sometimes mixed selective?

In this work, we develop a theory that precisely determines, for any dataset, whether the optimal learned representations will be modular or not. More precisely, in the linear autoencoder setting, we show that modularity in biologically constrained representations is governed by a *sufficient spread* condition that can roughly be thought of as measuring the extent to which the *range* of the source variables (sources, aka factors of variation) is *rectangular*. This sufficient spread condition bears resemblance to identifiability conditions derived in the matrix factorisation literature (Tatli

and Erdogan, 2021a;b), though both the precise form of the problem we study and the condition we derive differ (Appendix A). This condition on the sources is much weaker than the case of mutual independence studied in previous work (Whittington et al., 2023), and commensurately broadens the settings we can understand using our theory. For example, the loosening from statistical independence to rectangular support enables us to predict when linear recurrent neural network (RNN) representations of dynamic variables modularise.

Further, these results empirically generalise to nonlinear settings: we show that our source support conditions predict modularisation in nonlinear feedforward networks on supervised and autoencoding tasks as well as in nonlinear RNNs. We also fruitfully apply our theory to neuroscience data. We provide an explanation for why grid cells only sometimes warp in the presence of rewards, on the basis of the support independence properties of space and reward. Further, we use these results to suggest other settings in which mixed-selectivity might appear beyond the traditional nonlinear classification theory. In summary, our work contributes to the growing understanding of neural modularisation by highlighting how source support determines modularisation and explaining puzzling observations from both the brain and neural networks in a cohesive normative framework.

## 2 MODULARISATION IN BIOLOGICALLY INSPIRED LINEAR AUTOENCODERS

We begin with our main technical result: necessary and sufficient conditions for the modularisation of biologically inspired linear autoencoders.

### 2.1 PRELIMINARIES

Let $s \in \mathbb{R}^{d_s}$ be a vector of $d_s$ scalar source variables (sources, aka factors of variation). We are interested in how the empirical distribution, $p(s)$, affects whether the sources' neural encoding (latents), $z \in \mathbb{R}^{d_z}$, are *modular* with respect to the sources, i.e., whether each neuron (latent) is functionally dependent on at most one source. Following Whittington et al. (2023), we build a simplified model in which neural firing rates perfectly linearly autoencode the sources while maintaining nonnegativity (since firing rates are nonnegative):

$$z = W_{\text{in}} s + b_{\text{in}}, \quad s = W_{\text{out}} z + b_{\text{out}}, \quad z \geq 0. \tag{1}$$

Subject to the above constraints, we study the representation that uses the least energy, as in the classic efficient coding hypothesis (Barlow et al., 1961). We quantify this using the $\ell^2$ norm of the firing rates and weights (other activity norms considered later):

$$\min_{W_{\text{in}}, b_{\text{in}}, W_{\text{out}}, b_{\text{out}}} \left\langle ||z||_2^2 \right\rangle_{p(s)} + \lambda \left( ||W_{\text{out}}||_F^2 + ||W_{\text{in}}||_F^2 \right) \text{ s.t. (1).} \tag{2}$$

We remark that there are common analogues of representational nonnegativity and weight energy efficiency in modern machine learning (ReLU activation functions and weight decay, respectively). When the sources are statistically independent, i.e. $p(s) = \prod_{i=1}^{d_s} p(s_i)$, then the optima of (2) have modular latents (Whittington et al., 2023). We now improve on this result by showing necessary and sufficient conditions that guarantee modularisation for any dataset, not just those that have statistically independent sources.

### 2.2 INTUITION FOR SOURCE SUPPORT MODULARISATION CONDITIONS

To provide intuition, consider a hypothetical mixed selective neuron,

$$z_j = w_{j1} s_1 + w_{j2} s_2 + b_j, \tag{3}$$

It is functionally dependent on both $s_1$ and $s_2$, i.e. it is mixed selective. Perhaps, however, a modular representation, in which this neuron is broken into two separate modular encodings, would be better. We can create such a solution with two neurons each coding a single source:

$$z_{j'} = w_{j1} s_1 + b_{j'}, \quad z_{j''} = w_{j2} s_2 + b_{j''}. \tag{4}$$

For simplicity, we assume the two sources are linearly uncorrelated, have mean zero, and are supported on an interval centered at zero (Figure 1a top row; we relax these assumptions in our full theory). Then, for fixed $w_{j1}$ and $w_{j2}$, the optimal (energy efficient) bias should be large enough to keep the representation nonnegative, but no larger:

$$b_j = - \min_{s_1, s_2} [w_{j1} s_1 + w_{j2} s_2], \quad b_{j'} = - \min_{s_1} w_{j1} s_1, \quad b_{j''} = - \min_{s_2} w_{j2} s_2, \tag{5}$$

where the minimisations are over the empirical distribution $p(s_1, s_2)$. Now that both representations are specified, we can compare their costs. In our problem modularity is driven only by the activity loss (Appendix B). Further, in this simplified setting, most terms in the activity loss are zero or cancel, and one finds that the mixed selective (3) case uses less energy than the modular (4) when

$$b_j^2 < b_{j'}^2 + b_{j''}^2. \tag{6}$$

The key takeaway lies in how $b_j$ is determined by a joint minimization over $s_1$ and $s_2$ (5). Assume positive $w_{j1}$ and $w_{j2}$; then if $s_1$ and $s_2$ take their minima simultaneously, as in the middle row of Figure 1a, the mixed bias must be large:

$$b_j = -\min_{s_1, s_2}[w_{j1}s_1 + w_{j2}s_2] = -\min_{s_1}[w_{j1}s_1] - \min_{s_2}[w_{j2}s_2] = b_{j'} + b_{j''} \tag{7}$$

In this case, the energy of the mixed solution will always be worse than the modular, since $b_j^2 = (b_{j'} + b_{j''})^2 > b_{j'}^2 + b_{j''}^2$. Alternatively, mixing will be preferred when $s_1$ and $s_2$ *do not* take on their most negative values at the same time, as in the bottom row of Figure 1a, since then $b_j$ can be smaller while maintaining positivity, and the corresponding energy saving satisfies the key inequality (6).

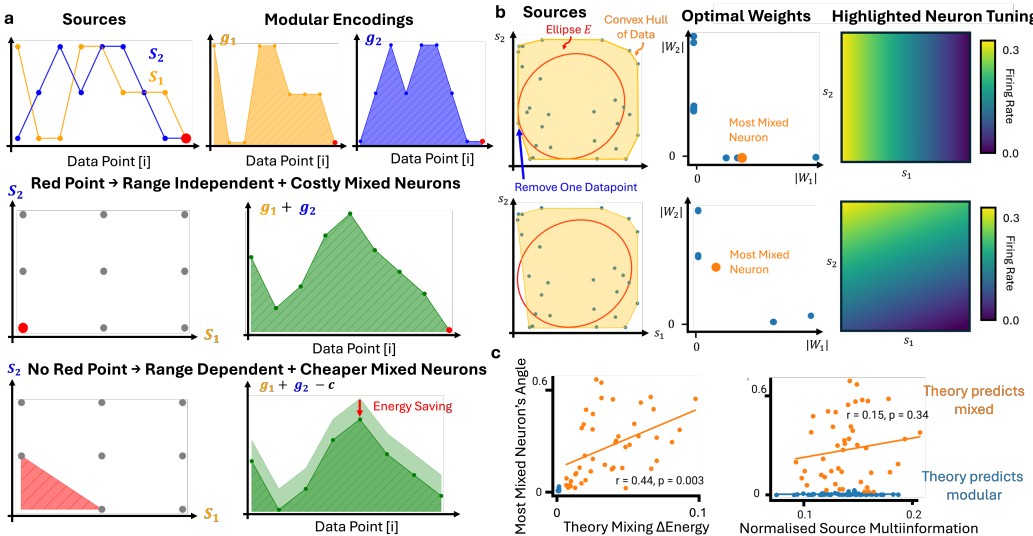

Figure 1: a) Top row: Values of two sources across a dataset and their modular encoding with associated costs (shaded regions). Middle row: If the dataset includes the red datapoint (left) then the two variables take their minimal value at the same time, and the mixed encoding must include a large bias, using more energy (right). Bottom row: If the red point is instead missing (left) then the mixed encoding can use a smaller bias while remaining positive and save energy (right). b) Our modularisation conditions (Theorem 2.1) are equivalent to whether the convex hull of the data (orange regions) encloses a data-derived ellipse (Theorem 2.2). The precision of these conditions is validated in two experiments (rows) in which the source support differs by a single datapoint. Removing this datapoint stops the convex hull from enclosing the key ellipse, resulting in a mixed-selective representation. In the middle column, we plot the weight vectors of the neurons. The top representation is modular as all weight vectors are axis aligned, whereas the bottom contains mixed selective neurons. This is reiterated by extracting the most mixed selective neuron (orange dot) and plotting its activity as a function of the sources (right column). c) Our conditions (left) predict modularisation better than measures of statistical independence such as source multiinformation (right). Here the y axis measures how mixed the representation is by the largest angle between a neuron's weight vector and one of the source axes, and the left x axis measures how much the inequalities are broken by - a rough heuristic for how mixed the optimal solution is.

## 2.3 Precise Conditions for Modularising Biologically Inspired Linear Autoencoders

We now make precise the intuition developed above. All proofs are deferred to Appendix B.

**Theorem 2.1.** *Let $s \in \mathbb{R}^{d_s}$, $z \in \mathbb{R}^{d_z}$, $W_{\text{in}} \in \mathbb{R}^{d_z \times d_s}$, $b_{\text{in}} \in \mathbb{R}^{d_z}$, $W_{\text{out}} \in \mathbb{R}^{d_s \times d_z}$, and $b_{\text{out}} \in \mathbb{R}^{d_s}$, with $d_z > d_s$. Consider the constrained optimization problem*

$$\min_{W_{\text{in}}, b_{\text{in}}, W_{\text{out}}, b_{\text{out}}} \quad \left\langle ||z^{[i]}||_2^2 \right\rangle_i + \lambda \left( ||W_{\text{in}}||_F^2 + ||W_{\text{out}}||_F^2 \right) \tag{8}$$
$$\text{s.t.} \quad z^{[i]} = W_{\text{in}} s^{[i]} + b_{\text{in}}, \ s^{[i]} = W_{\text{out}} z^{[i]} + b_{\text{out}}, \ z^{[i]} \geq 0,$$

*where $i$ indexes a finite set of samples of $s$. At the minima of this problem, the representation modularises, i.e. each row of $W_{\text{in}}$ has at most one non-zero entry, iff the following inequality is satisfied for all $w \in \mathbb{R}^{d_s}$:*

$$\left( \min_i [w^\top \bar{s}^{[i]}] \right)^2 + \sum_{j,j' \neq j}^{d_s} w_j w_{j'} \left\langle \bar{s}_j^{[i]} \bar{s}_{j'}^{[i]} \right\rangle_i > \sum_{j=1}^{d_s} \left( w_j \min_i \bar{s}_j^{[i]} \right)^2, \tag{9}$$

*where $\bar{s} := s - \left\langle s^{[i]} \right\rangle_i$ and assuming that $\left| \min_i \bar{s}_j^{[i]} \right| \leq \max_i \bar{s}_j^{[i]} \ \forall j \in [d_s]$ w.l.o.g.*

Our theory prescribes a set of inequalities (9) that determine whether an optimal representation is modular. These inequalities come from the difference in activity energy between a modular and mixed solutions, just like the intuition we built up in Section 2.2, and in particular (6). If a single inequality is broken, the optimal representation is mixed (at least in part); else, it is modular: optimally, each neuron's activity is a function of a single source. These inequalities depend on two key properties of the sources: the shape of the source distribution's support in extremal regions, and the pairwise source correlations. Remarkably, they do not depend on the energy tradeoff parameter $\lambda$. These inequalities can be visualised, as done in the wrapped figure below. For a given dataset, $\{s^{[i]}\}$ (blue dots), we can calculate $\left\langle \bar{s}_j^{[i]} \bar{s}_{j'}^{[i]} \right\rangle_i$ and each $\min_i \bar{s}_j^{[i]}$ as they are simple functions of the dataset. For all unit $w$, we can draw the line

$$w^\top x = \sqrt{ \sum_{j=1}^{d_s} \left( w_j \min_i \bar{s}_j^{[i]} \right)^2 - \sum_{j,j' \neq j}^{d_s} w_j w_{j'} \left\langle \bar{s}_j^{[i]} \bar{s}_{j'}^{[i]} \right\rangle_i } = \sqrt{w^T F w}, \tag{10}$$

where we have defined the following matrix:

$$F = \begin{bmatrix} (\min_i s_1^{[i]})^2 & -\langle s_1^{[i]} s_2^{[i]} \rangle_i & -\langle s_1^{[i]} s_3^{[i]} \rangle_i & \cdots \\ -\langle s_1^{[i]} s_2^{[i]} \rangle_i & (\min_i s_2^{[i]})^2 & -\langle s_2^{[i]} s_3^{[i]} \rangle_i & \cdots \\ -\langle s_1^{[i]} s_3^{[i]} \rangle_i & -\langle s_2^{[i]} s_3^{[i]} \rangle_i & (\min_i s_3^{[i]})^2 & \cdots \\ \vdots & \vdots & \vdots & \ddots \end{bmatrix} \tag{11}$$

Each $w$ gives us a line, and if there is at least one line that bounds the source support (such as the red one), then an inequality is broken and the optimal representation is mixed. Else it will be modular. This exercise also motivates the following equivalent statement of our conditions.

**Theorem 2.2.** *In the same setting as Theorem 2.1, define the set $E = \{y : y^T F^{-1} y = 1\}$. Then an equivalent statement of Theorem 2.1 is the representation modularises iff $E$ lies within the convex hull of the datapoints.*

This therefore provides a simple test: create $F$ and draw the set $E$ which, if $F$ is positive definite (as it often is), is an ellipse as shown in the wrapped figure above. The optimal representation modularises iff the convex hull of the datapoints encloses $E$, Figure 1b. If $F$ has some positive and some negative eigenvalues, then the set $E$ is unbounded, and the optimal representation must mix.

## 2.4 RANGE INDEPENDENT VARIABLES MODULARISE

To clarify our result we present a particularly clean special case.

**Corollary 2.3.** *In the same setting as Theorem 2.1 the optimal representation modularises if all sources are pairwise extreme-point independent, i.e. if for all $j, j' \in [d_s]^2$:*

$$\min_i \left[ s_j^{[i]} \middle| s_{j'}^{[i]} \in \left\{ \max_{i'} s_{j'}^{[i']}, \min_{i'} s_{j'}^{[i']} \right\} \right] = \min_i s_j^{[i]}. \tag{12}$$

In other words, if the joint distribution is supported on all extremal corners, the optimal representation modularises.

As presented, our theory has some limitations. It focuses on sources that are 1-dimensional, and uses a specific choice of activity norm, the L2, when others might be more reasonable. In fact, however, the core result that it is the independence of the variables' range (i.e. how rectangular they are) that drives modularity is very broadly true. In Appendix B.4 we show that this core result generalises to other activity norms, and in Appendix B.5 we show that it also applies to the modularisation of multi-dimensional variables such as angles on a 2D circle. Thus, our key takeaway is that the independence of variables' range is what determines whether the optimal representation is modular.

## 2.5 EMPIRICAL TESTS

**Validation of linear autoencoder theory.** We show our inequalities correctly predict modularisation. In particular, as an illustration of the precision of our theory, we create a dataset which transitions from inducing modularising to mixing via the removal of a single critical datapoint (Figure 1b). Further, we generate many datasets, create the optimal representation, and measure the angle $\theta$ between the most-mixed neuron's weight vector and its closest source direction, a proxy for modularity. The left of Figure 1c shows that our theory correctly predicts which datasets are modular ($\theta = 0$). Further, despite our theory being binary (will it modularise or not?), empirically we see that the degree to which the inequalities in Theorem 2.1 are broken is a good proxy for how mixed the optimal representation is. Finally, on the right of Figure 1c we show that on the same datasets a measure of source statistical interdependence, as used in previous work, does not predict modularisation.

**Predictions beyond our theory.** From our theory we extract qualitative trends to empirically test in more complex settings. Since extremal points play an outsized role in determining modularisation, we consider three trends that highlight these effects. (1) Datasets from which successively smaller corners have been removed should become successively less mixed, until at a critical threshold the representation modularises. (2) It is vital that it is not just any data but corner slices that are removed. Removing similar amounts of random or centrally located data from the dataset should not cause as much mixing. (3) Introducing correlations into a dataset while preserving extreme-point or range independence should preserve modularity relatively well.

## 3 MODULARISATION IN BIOLOGICALLY INSPIRED NONLINEAR FEEDFORWARD NETWORKS

Motivated by our linear theoretical results, we explore how closely biologically constrained nonlinear networks match our predicted trends. We study nonlinear representations with linear and nonlinear decoding in supervised and unsupervised settings. Surprisingly, coarse properties predicted by our linear theory generalise empirically to these nonlinear settings (Figure 2).

**Metrics for representational modularity and inter-source statistical dependence.** To quantify the modularity of a nonlinear representation, we design a family of metrics called conditional information-theoretic modularity (CInfoM), an extension of the InfoM metric proposed by Hsu et al. (2023). Intuitively, a representation is modular if each neuron is informative about only a single source. We therefore calculate the conditional mutual information between a neuron's activity and each source variable given all other sources. The conditioning on other sources is necessary to remove the effect of sources leaking information about each other; prior works consider independent sources or do not account for this effect. CInfoM then measures the extent to which a neuron specialises to its favourite source, relative to its informativeness about all sources. In order to compare multiple schemes that change $p(\boldsymbol{s})$ in different ways, we report normalised source multiinformation (NSI) as a measure of source statistical interdependence. NSI involves estimating the source multiinformation (aka total correlation) $D_{\mathrm{KL}}(p(\boldsymbol{s}) \parallel \prod_{i=1}^{d_s} p(s_i))$ and normalizing by $\sum_{i=1}^{d_s} H(s_i) - \max_i H(s_i)$. We defer further exposition of the CInfoM metric to Appendix E.

**What-where regression.** Inspired by the modularisation of what and where into the ventral and dorsal visual streams, we present nonlinear networks with simple binary images in which one pixel is on. The network is trained to concurrently report within which region of the image ("where"), and where within that region ("what") the on pixel is found, producing two outputs which we take as our sources, each an integer between one and nine. (More complex inputs or one-hot labels also work,

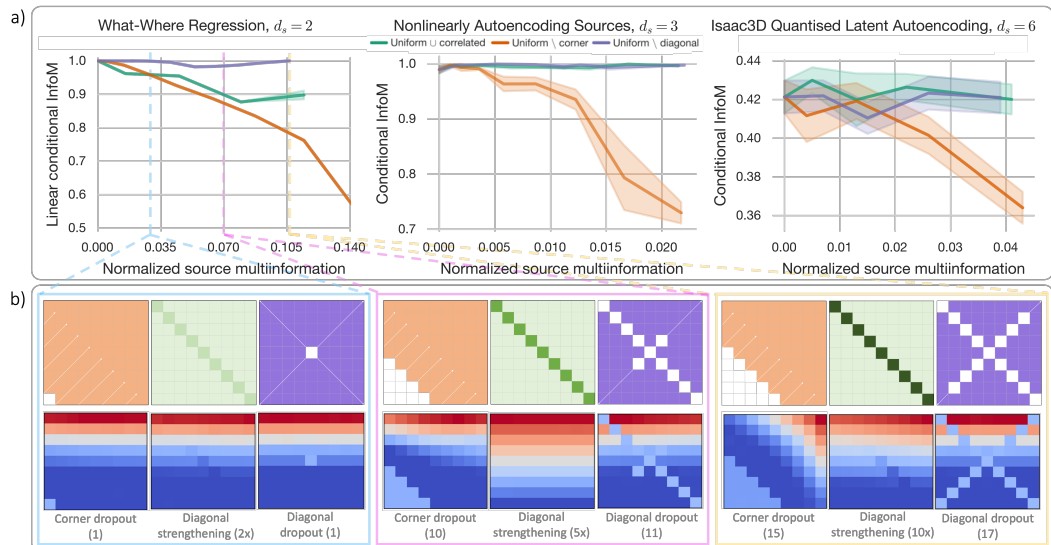

Figure 2: a) We study three tasks of varying source dimensionality $d_s$, model complexity, supervision signal, and input data modality (Section 3). For each task, we plot conditional InfoM, a measure of modularity, against normalised source multiinformation (NSI), a measure of the statistical dependencies in the sources. Across the three tasks, we observe that cutting corners severely degrades modularity (orange), whereas equivalently changing the NSI in ways that preserve the rectangular support boundary (green, purple) affects modularity less. Shaded regions denote standard error of the mean. b) For What-Where Regression, we visualise the source distributions at various NSI values (top) as well as the tuning of an example neuron. The $x$ ($y$) axes correspond to different what (where) values, and the colour shows the neuron's response. All neurons begin modular (left three plots), dropping corners heavily degrades modularity (left plot of subsequent groups of three).

see Appendix F.) We regularise the activity and weight energies, and enforce nonnegativity using a ReLU. If what and where are independent from one another, e.g. both uniformly distributed, then under our biological constraints (but not without them, see Appendix F) the hidden layer modularises what from where. Breaking the independence of what and where leads to mixed representations in patterns that qualitatively agree with our theory (Figure 2a left): cutting corners from the source support causes increasing mixing. Conversely, making other more drastic changes to the support, such as removing the diagonal, does not induce mixing. Similarly, introducing source correlations while preserving the rectangular support introduces less mixing when compared to corner cutting that induces the same amount of mutual information between what and where. The various source distributions at different NSI values are visualised in Figure 2b.

**Nonlinear autoencoding of sources.** Next we study a nonlinear autoencoder trained to autoencode a set of source variables. Again we find (Figure 2a middle) that under biological constraints independent source variables (NSI = 0) result in modular latents and that source corner cutting (orange) induces far more latent mixing compared to introducing source correlations while preserving rectangular support (green), or removing central data (purple).

**Disentangled representation learning of images.** Finally, for an experiment involving high-dimensional image data, we turn to a recently introduced state-of-the-art disentangling method, quantised latent autoencoding (QLAE; Hsu et al. (2023; 2024)). QLAE is the natural machine learning analogue to our biological constraints. It has two components: (1) the weights in QLAE are heavily regularised, like our weight energy loss, and (2) the latent space is axis-wise quantised, introducing a privileged basis with low channel capacity. In our biologically inspired networks, non-negativity and activity regularisation conspire to similarly structure the representation: nonnegativity creates a preferred basis, and activity regularisation encourages the representation to use as small a portion of the space as possible. We study the performance of QLAE trained to autoencode a subset of the Isaac3D dataset (Nie, 2019), a naturalistic image dataset with well defined underlying latent dimensions. We find the same qualitative patterns: corner cutting is a more important determinant of mixing than range-independent correlations or the removal of centrally located data (Figure 2a right).

# 4 MODULARISATION IN BIOLOGICALLY INSPIRED RECURRENT NETWORKS

Compared to feedforward networks, recurrent neural networks (RNNs) are often a much more natural setting for neuroscience. Excitingly, the core ideas of our analysis for linear autoencoders also apply to recurrent dynamical formulations, and similarly extend to experiments with nonlinear networks.

## 4.1 LINEAR RNNS

**Linear sinusoidal regression.** Linear dynamical systems can only autonomously implement exponentially growing, decaying, or stable sinusoidal functions. We therefore study linear RNNs with biological constraints trained to model stable sinusoidal signals at certain frequencies:

$$\boldsymbol{z}(t + \delta t) = \boldsymbol{W}_{\text{rec}}\boldsymbol{z}(t) + \boldsymbol{b}_{\text{rec}}, \quad \boldsymbol{W}_{\text{out}}\boldsymbol{z}(t) = \begin{bmatrix} \cos(\omega_1 t + \phi_1) \\ \cos(\omega_2 t + \phi_2) \end{bmatrix}, \quad \boldsymbol{z}(t) \geq \boldsymbol{0}. \quad (13)$$

We study the optimal nonnegative, efficient, recurrent representations, $\boldsymbol{z}_t$, and show that whether the representations of the two frequencies within $\boldsymbol{z}_t$ modularises depends on their ratio. We prove and verify empirically that if one frequency is an integer multiple of the other the encodings mix, whereas if their ratio is irrational they should modularise. Further, we show empirically, and prove in limited settings, that rational non-harmonic ratios should modularise (Appendix D). The intuition for this result is much the same as the linear autoencoding setting: the natural notions of sources are the signals $(\cos(\omega_1 t), \sin(\omega_1 t), \cos(\omega_2 t), \sin(\omega_2 t))$. Using these sources, we must simply ask: does their support allow for a reduction in activity energy via mixing? In Figure 3a, we visualise the source support for the three prototypical relationships between $\omega_1$ and $\omega_2$: irrational, rational (but not harmonic), and harmonic. Results of neural network verifications are in Figure 3b (details Appendix H). In the irrational case, the source support is essentially rectangular, so the model modularises. In the harmonic case, large chunks of various corners are missing from the source support, so the model mixes. In the rational case, even though the source support is quite sparse, the corners are sufficiently present such that modularising is still optimal.

**Modularisation of grid cells.** We now show that these spectral ideas can explain modules of grid cells. Grid cells are neurons in the mammalian entorhinal cortex that fire in a hexagonal lattice of positions (Hafting et al., 2005). They come in groups, called modules; grid cells within the same module have receptive fields (firing patterns) that are translated copies of the same lattice, and different modules are defined by their different lattice (Stensola et al., 2012) (for further literature review see Appendix A). Current theories suggest that the grid cell system can be modelled as a RNN with activations built from linear combinations of frequencies (Dorrell et al., 2023), just like the linear RNNs considered here. Importantly, these grid cell theories use the same biological constraints as in this work and show that modules form because the optimal code contains non-harmonically related frequencies that are encoded in different neurons (Figure 3e top). This modularisation can now be theoretically justified in our framework, since non-harmonic frequencies are range-independent and so should be modularised (Figure 3e bottom).

## 4.2 NONLINEAR RNNS

**Mixed sinusoidal regression for nonlinear RNNs.** To test how our ideas generalise beyond linear networks, we train nonlinear ReLU RNNs with biologically inspired constraints to perform a frequency mixing task (details Appendix H). We provide a pulse input $P_\omega(t) = \mathbb{I}\left[\text{mod}_\omega(t) = 0\right]$ at two frequencies, and the network has to output the resulting "beats" and "carrier" signals:

$$\boldsymbol{z}(t + \Delta t) = \text{ReLU}\left(\boldsymbol{W}_{\text{rec}}\boldsymbol{z}(t) + \boldsymbol{W}_{\text{in}}\begin{bmatrix} P_{\omega_1}(t) \\ P_{\omega_2}(t) \end{bmatrix} + \boldsymbol{b}_{\text{rec}}\right), \quad \begin{bmatrix} \cos([\omega_1 - \omega_2]t) \\ \cos([\omega_1 + \omega_2]t) \end{bmatrix} = \boldsymbol{W}_{\text{out}}\boldsymbol{z}_t + \boldsymbol{b}_{\text{out}}. \quad (14)$$

Results are in Figure 3c. Identical range-dependence properties but applied to the frequencies $\omega_1 - \omega_2$ and $\omega_1 + \omega_2$ determine whether or not the network modularises: irrational, range-independent frequencies modularise; harmonics, with their large missing corners, mix; and other rationally related frequencies are range-dependent but no sufficient corner is missing, so they modularise.

**Modularisation in nonlinear teacher-student distillation.** To test our predictions of when RNNs modularise, but in settings more realistic than pure frequencies, we generate training data trajectories from randomly initialised teacher RNNs with tanh activation function, and then train student RNNs

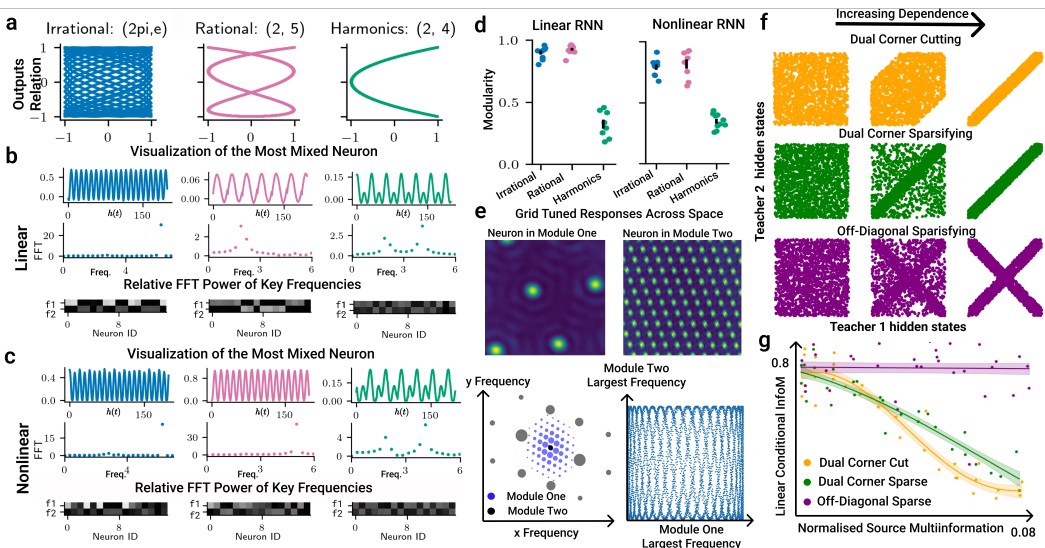

Figure 3: a) Source support visualisations: $\cos(\omega_1 t)$ vs. $\cos(\omega_2 t)$. b) The activity of the most mixed neuron in an optimal solution, the same neuron's Fourier spectrum, and the population's Fourier spectrum in the two task frequencies show clear modularisation in the irrational and rational cases, and clear mixing the in the harmonic case. c) Same as (b), but with *nonlinear* RNNs. d) Modularity of RNNs trained on 10 frequency pairs. e) Top: Two optimal grid cells from Dorrell et al. (2023) in different modules. Bottom: The modularisation of these two lattices can be understood from the range-independence of their constituent frequencies, as shown in the joint distribution of the most significant frequency in each module. f) Plot of joint distribution of hidden state activity for two teacher RNNs neurons as we increase the dependence in three different ways. g) Trends of modularity scores of student RNN in three different cases qualitatively agree with our theory.

(with a ReLU activation function) on these trajectories. The student's representation is constrained to be nonnegative (via its ReLU) and has its activity and weights regularised (see Appendix H.2 for details). Using carefully chosen inputs at each timestep, we are able to precisely control the teacher RNN hidden state activity (i.e., the source distribution). This allows us to change correlations/statistical independence of the hidden states, while either maintaining or breaking range independence. We consider three settings (Figure 3f). First, when statistical and range independence are progressively broken (in orange). Second, where statistical, and to a lesser extent range, independence gets progressively broken (in green). Third, where only statistical independence gets progressively broken (in purple). We observe that, in line with our theory, preserving the corner points preserves modularity (Figure 3g purple), while removing them breaks it (orange). Further, spasifying the corners breaks modularity, but more slowly than removing corners (green).

## 5 MODULAR OR MIXED CODES FOR SPACE & REWARD

We now apply our results to neuroscience to understand a puzzling difference in two seemingly similar recordings from entorhinal cortex. This brain area has been thought to contain precisely modular neurons, such as the grid cell code for self-position (Hafting et al., 2005), object vector cells that fire at a particular displacement from objects (Høydal et al., 2019), and heading direction cells (Taube, 2007). Two recent papers examined the influence of rewarded locations on the grid cell code and find differing effects on the modularity of grid cells. Butler et al. (2019) find that rewards rotate the grid cells, but preserve their pure spatial coding, while Boccara et al. (2019) find that the grid cells warp towards the rewards, becoming mixed-selective to reward and space.

Whittington et al. (2023) study this discrepancy and point to the importance of the reward distribution in these two tasks: Boccara et al. (2019) fix the positions of the possible rewards during one day, whereas Butler et al. (2019) alternate the animals between periods of free-foraging for randomly placed rewards and periods of specific rewarded locations. However, the arguments of Whittington et al. (2023), which rely on source independence, are insufficient to explain these modularisation effects, as in neither case are reward and position independent; even in the experiments of Butler et al. (2019) there are regions of space that are much more likely to be rewarded.

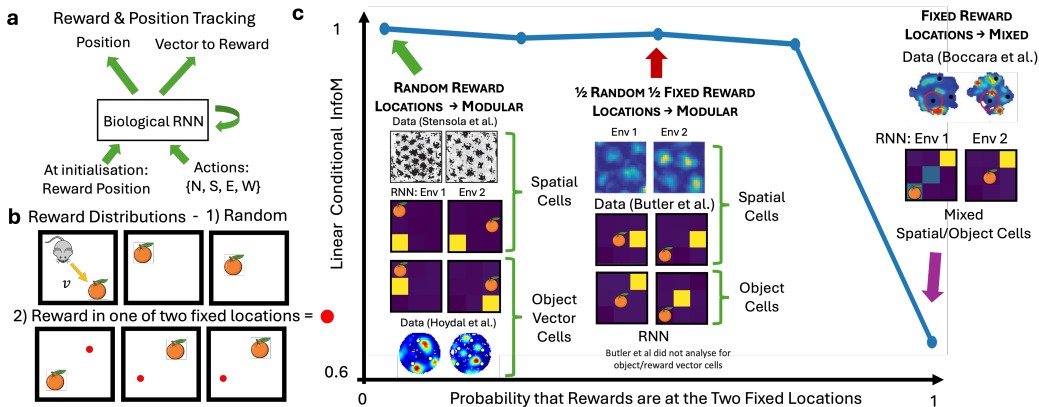

Figure 4: a, b) We train linear RNNs to integrate a sequence of actions to report their current position and displacement from a reward. In some proportion of the rooms, rewards are randomly scattered; in the rest, they are in one of two fixed locations. c) Matching the theory, if some of the rooms are random then reward and position are range-independent, and their representation modularises, as in Stensola et al. (2012) or Butler et al. (2019); if in all rooms the rewards occur in one of two fixed locations the representation mixes, with neurons encoding both reward and self position, as in Boccara et al. (2019).

However, with our improved understanding of modularisation, this makes sense. Despite Butler et al. (2019) correlating reward and position, critically, they leave them *range independent*—all combinations of reward and position are possible. On the other hand, Boccara et al. (2019) make certain combinations of reward and position impossible, making them not just correlated, but *range correlated*. As such, our theory matches this modularisation pattern. To test this we train a linear RNN with biological constraints (details Appendix H.3) to report both its self-position and displacement from a reward as it moves around an environment (Figure 4). We train RNNs on settings with different relationships between reward and position; for some RNNs the rewards are in fixed positions in every environment (range dependent and statistically dependent), for others reward and position are uniformly random in each room (range independent and statistically independent), and a final group experiences both settings in different proportions (range independent but statistically dependent). As we vary the proportion of fixed rooms, we find the optimal representation stays modular, containing separate spatial and reward-vector cells as in Butler et al. (2019), until all the sampled rooms have fixed reward positions, at which point neurons become mixed selective to reward and self-position, as in Boccara et al. (2019). As such, our theory now covers all known grid cell modularity results.

**Missed sources and mixed selectivity.** In contrast to the beautifully modular neurons in entorhinal cortex, recent work has highlighted neurons with mixed tuning to multiple navigational variables, such as position, heading direction, or speed (Hardcastle et al., 2017). We use our theory to highlight a potential artifact of this type of analysis. A purportedly mixed selective neuron may (in part) be purely selective for another unanalysed variable that is itself correlated with the measured spatial variables or behaviour. We highlight a simple example of this effect in Figure 5. Imagine a mouse is keeping track of three objects as it moves in an environment (Figure 5a); we model this as a linear RNN with biological constraints that must report the displacement of all objects from itself. We make the object positions range independent but correlated. As predicted by our theory, the RNN modularises these range-independent variables such that each neuron only encodes one object (Figure 5c). However, if an experimenter were only aware of two of the three objects, they would instead analyse the neural tuning with respect to those two objects, without being able to condition on the third. Due to the statistical dependencies between object positions, they would find mixed selective neurons that are in reality purely coding for the missing object (Figure 5d). We do not believe that all neurons in entorhinal cortex are modular; for example, there is clear evidence that neurons co-tuned to position and velocity are meaningfully mixed-selective, implementing the path-integration updates in the grid cell system (Vollan et al., 2025). Indeed, many of the neurons analysed in Hardcastle et al. (2017) *likely are* mixed selective. However, it is also seems likely that in more exploratory analyses such as these, mixed selectivity might often arise from missing encoded variables in the analysis.

Figure 5: a, b) We train linear RNNs to report displacement to three objects as an agent moves within many rooms. In each room, object instances are either completely random (a, top) or clustered in a random line (a, bottom); object positions are therefore range independent but correlated. c) The neurons are modular: each neuron's activity is conditionally informative of only a single object given the others. d) However, if an experimenter were only aware of two of the three objects, the neurons that purely encode the disregarded object would appear mixed selective due to the statistical dependencies between objects.

## 6 DISCUSSION

We present precise constraints on source distributions that cause linear feedforward and recurrent networks with biological constraints to modularise. These constraints depend on the co-range properties of variables, matching patterns of modularisation in both nonlinear networks and the brain.

**Limitations & future work.** That said, there remain many aspects of our theory that need improvement. Deriving necessary (rather than just sufficient) modularisation conditions for other norms and of multidimensional sources, extending to nonlinear settings, or to understand a less binarised notion of modularity (rather than perfectly modular or not), or developing a theory of linearly mixed sources as in matrix factorsiation work, are all attractive directions. Further, we assume perfect reconstruction. This leads us to some of our suprising conclusions, like how it is the support of the distribution that matters, even if some parts of the support have very low probability. In Appendix C we show that allowing the representation to tradeoff reconstruction accuracy with the other losses leads it to ignore very low probability points. Studying this tradeoff further would be interesting. Biologically, beyond the clear need for experimental validation, there are many pertinent effects whose impact on modularity would be interesting to study, such as connection sparsity, anatomical constraints, noise, or requiring additional computational roles from the network.

**Mixed selectivity.** In neuroscience our findings add nuance to the ongoing debate over mixed vs. modular neural coding. Theories of *nonlinear* mixed selectivity have argued that, analogously to a nonlinear kernel, such schemes permit linear readouts to decode nonlinear functions of the sources (Rigotti et al., 2013). This is likely a key part of the mixed selectivity found in some brain areas, like the cerebellum (Lanore et al., 2021), mushroom body (Aso et al., 2014), and perhaps certain prefrontal or hippocampal representations (Bernardi et al., 2020; Boyle et al., 2024). However, our theory raises the possibility for other explanations of both nonlinear and linear mixed selectivity that do *not* require tasks with nonlinear functions of the sources. First, purely from energy efficiency, our work analytically shows when range-dependent variables should be encoded in linearly mixed-selective representations, and empirically shows the same ideas apply to nonlinear mixed-selective representations, Figure 2. Further, since in our theory even statistically dependent variables might be represented modularly, we show that this causes problems for analysis techniques that do not account for all the variables encoded in a population, leading to spurious mixed selectivity, Figure 5. Future work could usefully develop principled approaches to distinguishing these options.

**Disentangling what?** We agree with Roth et al. (2023), who argue that range independence is the more meaningful form of independence for modularising variables. Further, the similarity between our conditions and those from the matrix factorisation literature (e.g. (Tatli and Erdogan, 2021a;b)), point to repeated instances of similar ideas. It is intriguing that constraints inspired by biology, which hold promise for disentangling (Whittington et al., 2023), and have links to state-of-the-art disentangling methods (Hsu et al., 2023; 2024), are sensitive to precisely this form of independence.

**Reproducibility Statement** Our theoretical work is fully detailed in the appendices, and code to reproduce our empirical work can be found at https://github.com/kylehkhsu/modular.

**Acknowledgements** The authors thank Basile Confavreux, Pierre Glaser, & Bariscan Bozkurt for conversations about proof techniques.

We thank the following funding sources: Gatsby Charitable Foundation to W.D., J.H.L. & P.E.L.; Sequoia Capital Stanford Graduate Fellowship and NSERC Postgraduate Scholarship – Doctoral to K.H.; Wellcome Trust (110114/Z/15/Z) to P.E.L.; Wellcome Trust (219627/Z/19/Z) to J.H.L.; Sir Henry Wellcome Postdoctoral Fellowship (222817/Z/21/Z) to JCRW.; Wellcome Principal Research Fellowship (219525/Z/19/Z), Wellcome Collaborator award (214314/Z/18/Z), and Jean-François and Marie-Laure de Clermont-Tonnerre Foundation award (JSMF220020372) to TEJB.; the Wellcome Centre for Integrative Neuroimaging is supported by core funding from the Wellcome Trust (203139/Z/16/Z).

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

## A    RELATED WORKS

**Machine learning.** The disentangled representation learning community has found inductive biases that lead networks to prefer modular or disentangled representations of nonlinear mixtures of independent variables. Examples include sparse temporal changes (Klindt et al., 2020), smoothness or sparseness assumptions in the generative model (Zheng et al., 2022; Horan et al., 2021), and axis-wise latent quantisation (Hsu et al., 2023; 2024). Relatedly, studies of tractable network models have identified a variety of structural aspects, in either the task or architecture, that lead to modularisation, including learning dynamics in gated linear networks (Saxe et al., 2022), architectural constraints in linear networks (Jarvis et al., 2023; Shi et al., 2022), compositional tasks in linear hypernetworks and nonlinear teacher-student frameworks (Schug et al., 2024; Lee et al., 2024), or task-sparsity in linear autoencoders (Elhage et al., 2022).

**Neuroscience.** Theories prompted by the recognition of mixed-selectivity in biological neurons have argued that nonlinearly mixed selective codes might exist to enable a linear readout to flexibly decode any required categorisation (Rigotti et al., 2013), suggesting a generalisability-flexibility tradeoff between modular and nonlinear mixed encodings (Bernardi et al., 2020). Modelling work has studied task-optimised network models of neural circuits, some of which have recovered mixed encodings (Nayebi et al., 2021). However, other models trained on a wide variety of cognitive tasks have found that networks contain meaningfully modularised components (Yang et al., 2019; Driscoll et al., 2024; Duncker et al., 2020). Dubreuil et al. (2022) study recurrent networks trained on similar cognitive tasks and find that, depending on the task structure, neurons in the trained networks can be grouped into populations defined by their connectivity patterns. To this developing body our work and its predecessor (Whittington et al., 2023) add understanding of when simple biological constraints would encourage encodings of many variables to be modularised into populations of single pure-tuned variables, or when mixed-selectivity would be preferred. Compared to (Whittington et al., 2023) which only works when source variables are statistically independent, our work determines whether the optimal representation is modular for any dataset even when source variables are not independent. This complete understanding of the boundary between mixed and modular codes is what allows us to understand behaviour in both biological and artificial neurons that previously eluded explanation.

**Independent components analysis and matrix factorization.** Our linear autoencoding problem (Theorem 2.1) bears some similarity to linear independent components analysis (ICA) and matrix factorization (MF). These assume data $y_i$ is linearly generated from a set of constrained sources, $y_i = As_i$, and build algorithms to recover $s_i$ from $y_i$. The literature studies different combinations of (1) assumptions on the sources $s_i$, e.g. independence (Bell and Sejnowski, 1995), nonnegativity(Plumbley, 2003), or boundedness Inan and Erdogan (2014); (2) assumptions on the mixing matrix $A$, e.g. nonnegative in nonnegative matrix factorisation and (3) algorithms, often deriving identifiability results: conditions that the data must satisfy for a given algorithm to succeed. Some work even derives biological implementations of such factorisation algorithms (Pehlevan et al., 2017; Bozkurt et al., 2022). In our work we study the conditions under which a particular biologically inspired representation modularises a set of bounded sources. We provide conditions on the sources under which the optimal representation is modular, morally similar to identifiability results (indeed, other identifiability results, like ours, sometimes take the form of sufficient spread conditions (Tatli and Erdogan, 2021a;b)). We differ from previous work in the representation we study (which has only been studied before by Whittington et al. (2023)), the form of our identifiability result (which is both necessary and sufficient, and has not been derived before), and how we apply these ideas, for example to neural data.

**Grid Cell Modules** One of the many surprising features of grid cells is their modular structure: each grid cell is a member of one module, of which there are a small number in rats. Grid cells from the same module have receptive fields that are translated versions of one another (Stensola et al., 2012). Previous work has built circuit models of such modules showing how they might path-integrate (Burak and Fiete, 2009), or has assumed the existence of multiple modules, then shown that they form a good code for space, and that parameter choices such as the module lengthscale ratio can be extracted from optimal coding arguments (Mathis et al., 2012; Wei et al., 2015). However, neither of these directions shows why, of all the ways to code space, a multi-modular structure is best. Dorrell et al. (2023) study a normative problem that explains the emergence of multiple modules grid cells as the best of all possible codes, but their arguments for the emergence of multiple grid modules are relatively intuitive. In this work we are able to formalise parts of their argument.

# B  POSITIVE LINEAR AUTOENCODER THEORY

In this section we derive our main mathematical results. We prove the main theorem, Appendix B.1, the equivalent statement in terms of the convex hull of the data, Appendix B.2, and the corollary on range-independent variables Appendix B.3. We then generalise this range-independent result to all activity norms, Appendix B.4, and to multidimensional variables, Appendix B.5.

## B.1  FULL THEORETICAL TREATMENT

Our proof of the main result, Theorem 2.1, takes the following strategy. First, we show that for a fixed encoding size, the weight loss is always minimised by orthogonalising, one example of which is modularising. Then we study when the activity loss is also minimised by modularising, and use that to tell us about the whole loss. Recall Theorem 2.1:

**Theorem B.1** (Main Result). *Let $s \in \mathbb{R}^{d_s}$, $z \in \mathbb{R}^{d_z}$, $W_{\text{in}} \in \mathbb{R}^{d_z \times d_s}$, $b_{\text{in}} \in \mathbb{R}^{d_z}$, $W_{\text{out}} \in \mathbb{R}^{d_s \times d_z}$, and $b_{\text{out}} \in \mathbb{R}^{d_s}$, with $d_z > d_s$. Consider the constrained optimization problem*

$$
\min_{W_{\text{in}}, b_{\text{in}}, W_{\text{out}}, b_{\text{out}}} \quad \left\langle ||z^{[i]}||_2^2 \right\rangle_i + \lambda \left( ||W_{\text{out}}||_F^2 + ||W_{\text{in}}||_F^2 \right)
$$
$$
\text{s.t.} \quad z^{[i]} = W_{\text{in}} s^{[i]} + b_{\text{in}}, \; s^{[i]} = W_{\text{out}} z^{[i]} + b_{\text{out}}, \; z^{[i]} \geq 0, \tag{15}
$$

*where $i$ indexes samples of $s$. At the minima of this problem, each row of $W_{\text{in}}$ has at most one non-zero entry, i.e. the representation modularises, iff the following inequality is satisfied for all $w \in \mathbb{R}^{d_s}$:*

$$
\left( \min_i [w^\top s^{[i]}] \right)^2 + \sum_{j,j' \neq j}^{d_s} w_j w_{j'} \left\langle \bar{s}_j^{[i]} \bar{s}_{j'}^{[i]} \right\rangle_i > \sum_{j=1}^{d_s} \left( w_j \min_i \bar{s}_j^{[i]} \right)^2, \tag{16}
$$

*where $\bar{s} := s - \left\langle s^{[i]} \right\rangle_i$ and assuming that $\left| \min_i \bar{s}_j^{[i]} \right| \leq \max_i \bar{s}_j^{[i]} \; \forall j \in [d_s]$ w.l.o.g.*

### B.1.1  WEIGHT LOSSES ARE MINIMISED BY MODULARISING

First, we show a lemma: that, for a fixed encoding size, the weight loss is minimised by orthogonalising the encoding of each source. Writing the representation as an affine function of the de-meaned sources:

$$
z^{[i]} = \sum_{j=1}^{d_s} u_j s_j^{[i]} + b = \sum_{j=1}^{d_s} u_j (s_j^{[i]} - \langle s_j^{[k]} \rangle_k) + b' = \sum_{j=1}^{d_s} u_j \bar{s}_j^{[i]} + b' \tag{17}
$$

then by fixed encoding size we mean that each $|u_j|$ is fixed. We express things in terms of mean-zero variables $\bar{s}_j^{[i]}$ since it is easier and we can freely shift by a constant offset.

**Theorem B.2** (Modularising Minimises Weight Loss). *The weight loss, $||W_{\text{out}}||_F^2 + ||W_{\text{in}}||_F^2$, is minimised, for fixed encoding magnitudes, $|u_j|$, by modularising the representation.*

*Proof.* First, we know that $W_{\text{in}}$ must be of the following form:

$$
W_{\text{in}} = [u_1 \quad \ldots \quad u_{d_s}] \tag{18}
$$

And therefore for a fixed encoding size the input weight loss is constant:

$$
||W_{\text{in}}||_F^2 = \text{Tr}[W_{\text{in}}^T W_{\text{in}}] = \sum_{j=1}^{d_s} |u_j|^2 \tag{19}
$$

So let's study the output weights, these are defined by the way they map the representation back to the sources:

$$
W_{\text{out}} \cdot z^{[i]} + b_{\text{out}} = \begin{bmatrix} s_1^{[i]} \\ \vdots \\ s_{d_s}^{[i]} \end{bmatrix} \tag{20}
$$

The min-norm $\boldsymbol{W}_{\text{out}}$ with this property is the Moore-Penrose Pseudoinverse, i.e. the matrix:

$$\boldsymbol{W}_{\text{out}} = \begin{bmatrix} \boldsymbol{U}_1^T \\ \vdots \\ \boldsymbol{U}_{d_s}^T \end{bmatrix} \tag{21}$$

Where each pseudoinverse $\boldsymbol{U}_i$ is defined by $\boldsymbol{U}_i^T \boldsymbol{u}_j = \delta_{ij}$ and lying completely within the span of the encoding vectors $\{\boldsymbol{u}_j\}$. We can calculate the norm of this matrix:

$$|\boldsymbol{W}_{\text{out}}|_F^2 = \text{Tr}[\boldsymbol{W}_{\text{out}} \boldsymbol{W}_{\text{out}}^T] = \sum_{j=1}^{d_s} |\boldsymbol{U}_j|^2 \tag{22}$$

Now, each of these capitalised pseudoinverse vectors must have some component along its corresponding lower case vector, and some component orthogonal to that:

$$\boldsymbol{U}_j = \frac{1}{|\boldsymbol{u}_j|^2} \boldsymbol{u}_j + \boldsymbol{u}_{j,\perp} \tag{23}$$

We've chosen the size of the component along $\boldsymbol{u}_j$ such that $\boldsymbol{U}_j^T \boldsymbol{u}_j = 1$, and the $\boldsymbol{u}_{j,\perp}$ is chosen so that $\boldsymbol{U}_j^T \boldsymbol{u}_k = \delta_{jk}$. Now, for a fixed size of $|\boldsymbol{u}_j|$, this sets a lower bound on the size of the weight matrix:

$$|\boldsymbol{W}_{\text{out}}|_F^2 = \sum_{j=1}^{d_s} \frac{1}{|\boldsymbol{u}_j|^2} + |\boldsymbol{u}_{j,\perp}|^2 \geq \sum_{j=1}^{d_s} \frac{1}{|\boldsymbol{u}_j|^2} \tag{24}$$

And this lower bound is achieved whenever the $\{\boldsymbol{u}_j\}_{j=1}^{d_s}$ vectors are orthogonal to one another, since then $\boldsymbol{u}_{j,\perp} = 0$. Therefore, we see that, for a fixed size of encoding, the weight loss is minimised when the encoding vectors are orthogonal, and that is achieved when the code is modular. □

### B.1.2 ACTIVITY LOSS

Now we will turn to studying the activity loss, and find conditions under which a modular representation is better than a mixed one. We compare two representations, a mixed neuron:

$$z_n^{[i]} = \sum_{j=1}^{d_s} u_{nj} \bar{s}_j^{[i]} + \Delta_n = \sum_{j=1}^{d_s} u_{nj} \bar{s}_j^{[i]} - \min_i [\sum_{j=1}^{d_s} u_{nj} \bar{s}_j^{[i]}] \tag{25}$$

And another representation in which we break this neuron apart into its modular form, preserving the encoding size of each source:

$$\boldsymbol{z}_n^{[i]} = \begin{bmatrix} u_{n1} \bar{s}_1^{[i]} \\ \vdots \\ u_{nd_s} \bar{s}_{d_s}^{[i]} \end{bmatrix} - \begin{bmatrix} |u_{n1}| \min_j \bar{s}_1^{[j]} \\ \vdots \\ |u_{nd_s}| \min_j \bar{s}_{d_s}^{[i]} \end{bmatrix} \tag{26}$$

The activity loss difference between the modular and mixed representations is:

$$\sum_j u_{nj}^2 (\min_i \bar{s}_j^{[i]})^2 - (\min_i [\sum_{j=1}^{d_s} u_{nj} \bar{s}_j^{[i]}])^2 - \sum_{j=1,k \neq j}^{d_s} u_{nj} u_{nk} \langle \bar{s}_j^{[i]} \bar{s}_k^{[i]} \rangle_i \tag{27}$$

So the modular loss is smaller if, for all $\boldsymbol{u}_n \in \mathbb{R}^{d_s}$:

$$(\min_i [\sum_{j=1}^{d_s} u_{nj} \bar{s}_j^{[i]}])^2 > \sum_j u_{nj}^2 (\min_i \bar{s}_j^{[i]})^2 - \sum_{j,k \neq j} u_{nj} u_{nk} \langle \bar{s}_j^{[i]} \bar{s}_k^{[i]} \rangle_i \tag{28}$$

Further, we can see that if this equation holds for one vector, $\boldsymbol{u}$, it also holds for any scaled version of that vector, therefore we can equivalently require the inequality to be true only for unit norm vectors.

### B.1.3 COMBINATION OF WEIGHT AND ACTIVITY

So, if these inequalities are satisfied you can always decrease the loss by modularising a mixed neuron. Therefore, both the weight and activity loss are minimised at fixed encoding size by modularising, and this holds for all encoding sizes, therefore the optimal solution must be modular. Hence these conditions are sufficient for modular solutions to be optimal.

Conversely, if one of the inequalities is broken then the activity loss is not minimised at the modular solution. We can move towards the optimal mixed solution by creating a new mixed encoding neuron with infinitesimally low activity, and mixed in proportions given by the broken inequality. This neuron can be created by taking small pieces of the modular encoding of each source, in proportion to the mixing. Doing this will decrease the activity loss. Further, the weight loss is minimised at the modular solution, so, to first order, moving away from modular solution doesn't change it. Therefore, for each modular representation, we have found a mixed solution that is better, so the optimal solution is not modular.

## B.2 Equivalence of Convex Hull Formulation

In this section we prove Theorem 2.2, an equivalent formulation of the infinitely many inequalities in Theorem B.1 in terms of the convex hull, restated below:

**Theorem B.3.** *In the setting as Theorem B.1, define the following matrix:*

$$\boldsymbol{F} = \begin{bmatrix} (\min_i s_1^{[i]})^2 & -\langle s_1^{[i]} s_2^{[i]} \rangle_i & -\langle s_1^{[i]} s_3^{[i]} \rangle_i & \dots \\ -\langle s_1^{[i]} s_2^{[i]} \rangle_i & (\min_i s_2^{[i]})^2 & -\langle s_2^{[i]} s_3^{[i]} \rangle_i & \dots \\ -\langle s_1^{[i]} s_3^{[i]} \rangle_i & -\langle s_2^{[i]} s_3^{[i]} \rangle_i & (\min_i s_3^{[i]})^2 & \dots \\ \vdots & \vdots & \vdots & \ddots \end{bmatrix} \tag{29}$$

*Define the set $E = \{\boldsymbol{y} : \boldsymbol{y}^T \boldsymbol{F}^{-1} \boldsymbol{y} = 1\}$. Then an equivalent statement of the modularisation inequalities (16), is the representation modularises iff $E$ lies within the convex hull of the datapoints.*

Notice that the matrix $\boldsymbol{F}$ is not guaranteed to be positive definite, meaning $E$ is not necessarily an ellipse. In these cases no convex hull can contain the unbounded quadric it forms, and the optimal solution is mixed regardless of the particular distribution of the datapoints.

We'll begin by rewriting the inequalities, then we'll show each direction of the equivalence.

### B.2.1 Rewrite Inequality Constraints

The current form of the inequality is that, $\forall \boldsymbol{w} \in \mathbb{R}^n$:

$$(\min_i \boldsymbol{w}^T \boldsymbol{s}^{[i]})^2 + \sum_{j,j' \neq j}^{d_s} w_j w_{j'} \langle s_j^{[i]} s_{j'}^{[i]} \rangle_i \geq \sum_j w_j^2 (\min_i s_j^{[i]})^2 \tag{30}$$

If this equation holds for one vector, $\boldsymbol{w}$, it also holds for all positively scaled versions of the vector, $\boldsymbol{v} = \lambda \boldsymbol{w}$ for $\lambda > 0$, so we can equivalently state that this condition must only hold for unit vectors.

Construct the matrix $\boldsymbol{F}$ as above (29), then this condition can be rewritten:

$$(\min_i \boldsymbol{w}^T \boldsymbol{s}^{[i]})^2 \geq \boldsymbol{w}^T \boldsymbol{F} \boldsymbol{w} \quad \forall \boldsymbol{w} \in \mathbb{R}^n, |\boldsymbol{w}| = 1 \tag{31}$$

Further, notice that we can swap the minima in the inequality for a maxima. We can do this because if the inequality is satisfied for a particular value of $\boldsymbol{w}$ then the following inequality is satisfied for $-\boldsymbol{w}$:

$$(\max_i \boldsymbol{w}^T \boldsymbol{s}^{[i]})^2 \geq \boldsymbol{w}^T \boldsymbol{F} \boldsymbol{w} \quad \forall \boldsymbol{w} \in \mathbb{R}^n, |\boldsymbol{w}| = 1 \tag{32}$$

Since (31) is satisfied for all $\boldsymbol{w}$, and all $\boldsymbol{w}$ have their $-\boldsymbol{w}$ unit vector pairs, so is (32). Similarly, if (32) is satisfied for all unit vectors $\boldsymbol{w}$, (31) is satisfied but using $-\boldsymbol{w}$. Hence the two are equivalent.

### B.2.2 Sufficiency of Condition

First, let's show the convex hull enclosing $E$ is a sufficient condition. Choose an arbitrary unit vector $\boldsymbol{w}$. Choose the point within $E$ that maximises the dot product with $\boldsymbol{w}$:

$$\boldsymbol{y}_w = \arg\max_{\boldsymbol{y} \in E} \boldsymbol{w}^T \boldsymbol{y} \tag{33}$$

We can find $\boldsymbol{y}_w$ by lagrange optimisation under the constraint that $\boldsymbol{y}^T \boldsymbol{F}^{-1} \boldsymbol{y} = 1$:

$$\boldsymbol{y}_w = \frac{\boldsymbol{F} \boldsymbol{w}}{\sqrt{\boldsymbol{w}^T \boldsymbol{F} \boldsymbol{w}}} \tag{34}$$

By the definition of $E$ and its enclosure in the convex hull of the points we can write:

$$\boldsymbol{y}_w = \sum_i \lambda_i \boldsymbol{s}^{[i]} \quad \lambda_i > 0, \sum_i \lambda_i \leq 1 \tag{35}$$

Now:

$$\boldsymbol{w}^T \boldsymbol{y} = \sqrt{\boldsymbol{w}^T \boldsymbol{F} \boldsymbol{w}} = \sum_i \lambda_i \boldsymbol{w}^T \boldsymbol{s}^{[i]} \leq \max_i \boldsymbol{w}^T \boldsymbol{s}^{[i]} \sum_i \lambda_i = \max_i \boldsymbol{w}^T \boldsymbol{s}^{[i]} \tag{36}$$

Therefore, we get our our main result:

$$(\max_i \boldsymbol{w}^T \boldsymbol{s}^{[i]})^2 \geq \boldsymbol{w}^T \boldsymbol{F} \boldsymbol{w} \tag{37}$$

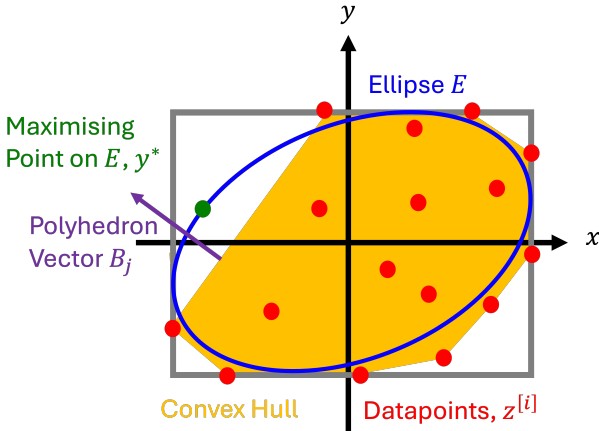

Figure 6: If the convex hull does not enclose the ellipse then you can always find a unit vector that breaks the inequality. In particular, this is true for the unit vector that defines the faces of the convex hull, $\boldsymbol{B}_j$.

### B.2.3   NECESSITY OF CONDITION

Now we will show the necessity of this condition. We will do this by showing that if one of the points in $E$ is not within the convex hull of the datapoints then we can find one of the inequalities in (32) that is broken.

First, get all the datapoints that for some unit vector $\boldsymbol{w}$ maximise $\boldsymbol{w}^T \boldsymbol{s}^{[i]}$ over the dataset. Construct the convex hull of these points, the polyhedron $P$. You can equivalently characterise any polyhedra by a set of linear inequalities: $P = \{\boldsymbol{x} \in \mathbb{R} : \boldsymbol{B}\boldsymbol{x} \leq \boldsymbol{b}\}$. Denote with $\boldsymbol{B}_j$ the jth row of $\boldsymbol{B}$. We can always choose to write the inequalities in such a way that the rows of $\boldsymbol{B}$ are unit length, by appropriately rescaling each $b_j$. Assume we have done this.

Assume the convex hull does not contain the entire ellipse. Therefore there is at least one point on the ellipse that breaks one of the inequalities defining the polyhedron. These points are charactersied by being both on the ellipse, $\boldsymbol{y}^T \boldsymbol{F}^{-1}\boldsymbol{y} = 1$, but at least one of the inequalities defining the polyhedron is broken: $\boldsymbol{B}_j^T \boldsymbol{y} > b_j$. Let's call the set of points on the ellipse for which this inequality is broken $\mathscr{B}_j = \{\boldsymbol{y} \in \mathbb{R}^n : \boldsymbol{y}^T \boldsymbol{F}^{-1}\boldsymbol{y} = 1 \quad \text{and} \quad \boldsymbol{B}_j^T \boldsymbol{y} > b_j\}$.

Now, choose the element of $\mathscr{B}_j$ that maximises $\boldsymbol{B}_j^T \boldsymbol{y}$, call it $\boldsymbol{y}^*$. This will also be the element of $E$ that maximises $\boldsymbol{B}_j^T \boldsymbol{y}$, which by the previous lagrange optimisation, (34), has the following dot product: $\boldsymbol{B}_j^T \boldsymbol{y}^* = \sqrt{\boldsymbol{B}_j^T \boldsymbol{F} \boldsymbol{B}_j}$.

But now we derive a contradiction. This point is outside the polyhedron, $\boldsymbol{B}_j^T \boldsymbol{y} > b_j$, whereas all points inside the polyhedron, including all the datapoints, satisfy all inequalities, hence, $\boldsymbol{B}_j^T \boldsymbol{s}^{[i]} \leq b_j \forall \boldsymbol{s}^{[i]}$. But, $\boldsymbol{B}_j^T \boldsymbol{y}^* = \sqrt{\boldsymbol{B}_j^T \boldsymbol{F} \boldsymbol{B}_j} > b_j$, therefore:

$$\boldsymbol{B}_j^T \boldsymbol{s}^{[i]} \leq b_j < \sqrt{\boldsymbol{B}_j^T \boldsymbol{F} \boldsymbol{B}_j} \quad \forall \boldsymbol{s}^{[i]} \tag{38}$$

And hence we've found a unit vector $\boldsymbol{B}_i$ such that:

$$\max_i \boldsymbol{B}_j^T \boldsymbol{s}^{[i]} < \sqrt{\boldsymbol{B}_j^T \boldsymbol{F} \boldsymbol{B}_j} \quad \rightarrow \quad (\max_i \boldsymbol{B}_j^T \boldsymbol{s}^{[i]})^2 < \boldsymbol{B}_j^T \boldsymbol{F} \boldsymbol{B}_j \tag{39}$$

Hence one of the inequalities in (32) is broken, showing the necessity.

### B.3 RANGE-INDEPENDENT VARIABLES

We will now prove the corollary on range dependence, Theorem 2.3, recapped here:

**Corollary B.4.** *In the same setting as Theorem B.1 the optimal representation modularises if all sources are pairwise extreme-point independent, i.e. if*

$$\min_i \left[ s_j^{[i]} \Big| s_{j'}^{[i]} \in \left\{ \max_{i'} s_{j'}^{[i']}, \min_{i'} s_{j'}^{[i']} \right\} \right] = \min_i s_j^{[i]} \tag{40}$$

*for all $j, j' \in [d_s]^2$.*

*Proof.* If the variables are extreme-point independent then we can take the min inside the sum in the definition of the mixed neuron bias:

$$z_n^{[i]} = \sum_{j=1}^{d_s} u_{nj} \bar{s}_j^{[i]} + \Delta_n = \sum_{j=1}^{d_s} u_{nj} \bar{s}_j^{[i]} - \min_i [\sum_{j=1}^{d_s} u_{nj} \bar{s}_j^{[i]}] = \sum_{j=1}^{d_s} \left( u_{nj} \bar{s}_j^{[i]} - \min_i [u_{nj} \bar{s}_j^{[i]}] \right) \tag{41}$$

Now we can compute the activity loss:

$$\langle (z_n^{[i]})^2 \rangle_i =$$

$$\sum_{j=1}^{d_s} \langle \left( u_{nj} \bar{s}_j^{[i]} - \min_i [u_{nj} \bar{s}_j^{[i]}] \right)^2 \rangle_i + \sum_{j,j'=1, j\neq j}^{d_s} \langle \left( u_{nj} \bar{s}_j^{[i]} - \min_i [u_{nj} \bar{s}_j^{[i]}] \right) \left( u_{nj} \bar{s}_{j'}^{[i]} - \min_i [u_{nj} \bar{s}_{j'}^{[i]}] \right) \rangle_i \tag{42}$$

The cross terms at the end are strictly positive, and each of the squared terms on the left is the same or larger than its corresponding modular neuron, since either $u_{nj}$ is positive and it is the same, or $u_{nj}$ is negative and it is greater than or equal to the equivalent modular neuron. Therefore the mixed energy cost is larger than the modular energy cost for extreme-point independent variables, so the optimal representation should modularise. $\square$

### B.4 EXTENSION TO ALL ACTIVITY NORMS

We now extend our theoretical results to other activity norms beyond L2. As shown in theorem Theorem B.2, the weight loss is minimised by orthogonalising the coding of different variables, so we can focus on the activity loss. We will show that, for all norms, modular solutions are optimal if the variables are range independent.

Consider a set of modular neurons:

$$\boldsymbol{z}_n^{[i]} = \begin{bmatrix} |u_{n1}|(\bar{s}_1^{[i]} - \min_j \bar{s}_1^{[j]}) \\ \vdots \\ |u_{nd_s}|(\bar{s}_{d_s}^{[i]} - \min_j \bar{s}_{d_s}^{[i]}) \end{bmatrix} \geq \boldsymbol{0} \tag{43}$$

And let's see if we could usefully mix these encodings together for any set of mixing coefficients $\boldsymbol{u}_n \in \mathbb{R}^{d_s}$:

$$z_n^{[i]} = \sum_{j=1}^{d_s} u_{nj}\bar{s}_j^{[i]} + \Delta_n = \sum_{j=1}^{d_s} u_{nj}\bar{s}_j^{[i]} - \min_i[\sum_{j=1}^{d_s} u_{nj}\bar{s}_j^{[i]}] \geq 0 \tag{44}$$

If the variables are range-independent then we can move the min inside the sum:

$$\Delta_n = -\min_i[\sum_{j=1}^{d_s} u_{nj}\bar{s}_j^{[i]}] = -\sum_{j=1}^{d_s} \min_i u_{nj}\bar{s}_j^{[i]} \tag{45}$$

Then the Lp norm is:

$$\langle|z_n^{[i]}|^p\rangle_i = \langle|\left(\sum_{j=1}^{d_s} u_{nj}\bar{s}_j^{[i]} + \Delta_n\right)|^p\rangle_i = \langle|\left(\sum_{j=1}^{d_s} u_{nj}\bar{s}_j^{[i]} - \min_i u_{nj}\bar{s}_j^{[i]}\right)|^p\rangle_i \tag{46}$$

Now using the convexity of the Lp norm:

$$\langle|\left(\sum_{j=1}^{d_s} u_{nj}\bar{s}_j^{[i]} - \min_i u_{nj}\bar{s}_j^{[i]}\right)|^p\rangle_i \geq \sum_{j=1}^{d_s} \langle|u_{nj}\bar{s}_j^{[i]} - \min_i u_{nj}\bar{s}_j^{[i]}|^p\rangle_i \tag{47}$$

with equality only if p is equal to 1, or if at most one of the entries of $\boldsymbol{u}$ are 0.

Finally, since we orient our variables such that $|\min_i \bar{s}_j^{[i]}| < \max_i \bar{s}^{[i]}$, without loss of generality,:

$$\langle|u_{nj}\bar{s}_j^{[i]} - \min_i u_{nj}\bar{s}_j^{[i]}|^p\rangle_i \geq \langle|u_{nj}|^p\langle|\bar{s}_j^{[i]} - \min_i \bar{s}_j^{[i]}|^p\rangle_i \tag{48}$$

So we get our final result that the mixed activity loss is greater than or equal to the modular activity loss if the variables are range independent:

$$\langle|z_n^{[i]}|^p\rangle_i \geq \sum_{j=1}^{d_s} |u_{nj}|^p\langle|\bar{s}_j^{[i]} - \min_i \bar{s}_j^{[i]}|^p\rangle_i = \langle|\boldsymbol{z}_n^{[i]}|^p\rangle_i \tag{49}$$

So, as long as $p > 1$, the activity loss is lowered by modularising the representation of range-independent variables. Further, since the weight loss is also minimised, the optimal solution will be modular.

Two caveats: first, in the particular case of the L1 norm all that we can guarantee is that the activity loss of a modular solution is equal to that of a mixed solution. This means that, even in the case of range-independent variables, there might be orthogonal but mixed solutions that are equally good as the modular solution. For $p > 1$ this caveat is not needed.

Second, this is a set of sufficient conditions, range-independent variables are optimally encoded modularly. There will be other nearly range-independent variables that are also optimally encoded modularly, just as we found for the L2 case.

B.5 EXTENSION TO MULTIDIMENSIONAL VARIABLES

Many variables in the real world, such as angles on a circle, are not 1-dimensional. Ideally our theory would tell us when such multi-dimensional sources were modularised. Unfortunately, precise necessary and sufficient conditions are not an easy step from our current theory because our proof strategies make extensive use of the optimal modular encoding. For one-dimensional sources this is very easy to find, simply align each source correctly in a neuron, and make the representation positive. Finding similar optimal modular solutions for multidimensional sources is much harder, potentially as hard as simply solving the optimisation problem.

However, we can make progress on a simpler problem: showing that range-independence is a sufficient condition for a set of multi-dimensional sources to modularise. Consider two multidimensional sources $\boldsymbol{x}^{[i]} \in \mathbb{R}^{d_x}$ and $\boldsymbol{y}^{[i]} \in \mathbb{R}^{d_y}$, and an encoding in which they are mixed in single neurons:

$$z_n^{[i]} = \sum_{j=1}^{d_x} u_{nj} x_j^{[i]} + \sum_{j=1}^{d_y} v_{nj} y_j^{[i]} + \Delta_n \tag{50}$$

Whatever choice of coefficient vectors, $\boldsymbol{u}$ and $\boldsymbol{v}$, or activity norm (as long as $p > 1$), we can consider breaking this mixed coding into two modular neurons. Define new variables $a^{[i]} = \sum_{j=1}^{d_x} u_{nj} x_j^{[i]}$ and $b^{[i]} = \sum_{j=1}^{d_y} v_{nj} y_j^{[i]}$. If the two multidimensional sources are range-independent, then so too are $a$ and $b$. Define the modular representation:

$$\boldsymbol{z}_n^{[i]} = \begin{bmatrix} a^{[i]} - \min_j a^{[j]} \\ b^{[i]} - \min_j b^{[j]} \end{bmatrix} \tag{51}$$

All of our previous arguments apply to argue that this modular encoding will be preferred if the multidimensional variables are range-independent.

We just have to show that the weight loss is also reduced by modularising multidimensional sources, a small extension of Theorem B.2, which we shall now show. Write a modular code as:

$$\boldsymbol{z}^{[i]} = \boldsymbol{U}\boldsymbol{x}^{[i]} - \min_j[\boldsymbol{U}\boldsymbol{x}^{[j]}] + \boldsymbol{V}\boldsymbol{y}^{[i]} - \min_j[\boldsymbol{V}\boldsymbol{y}^{[j]}] \tag{52}$$

With the encoding of $\boldsymbol{x}$ and $\boldsymbol{y}$ spanning two different spaces: $\boldsymbol{U}^T \boldsymbol{V} = \boldsymbol{0}$.

Then define the minimal L2 norm readout weight matrix using the pseudoinverse:

$$\boldsymbol{W}_{\text{out}} = \begin{bmatrix} \boldsymbol{U}^\dagger \\ \boldsymbol{V}^\dagger \end{bmatrix} \tag{53}$$

Such that $\boldsymbol{W}_{\text{out}} \boldsymbol{z}^{[i]} = \begin{bmatrix} \boldsymbol{x}^{[i]} \\ \boldsymbol{y}^{[i]} \end{bmatrix}$ up to some constant offset that can be removed by the bias term in the readout. Now consider mixing the representation of the two multidimensional sources using an orthogonal matrix that preserves the encoding size of each variable, $\boldsymbol{O}$:

$$\boldsymbol{z}^{[i]} = \boldsymbol{U}\boldsymbol{x}^{[i]} + \boldsymbol{O}\boldsymbol{V}\boldsymbol{y}^{[i]} \qquad \boldsymbol{W}_{\text{out}} = \boldsymbol{W}_{\text{out}} = \begin{bmatrix} \boldsymbol{U}^\dagger + \boldsymbol{A} \\ \boldsymbol{V}^\dagger \boldsymbol{O}^T + \boldsymbol{B} \end{bmatrix} \tag{54}$$

Then in the subspace spanned by the columns of $\boldsymbol{U}$ $\boldsymbol{W}_{\text{out}}$ must be the same as $\boldsymbol{U}^\dagger$. However, in order to offset the newly aligned matrix $\boldsymbol{O}\tilde{\boldsymbol{V}}$ there must be an additional component orthogonal, $\boldsymbol{A}$:

$$\boldsymbol{W}_{out}\boldsymbol{U} = \boldsymbol{U}^\dagger \boldsymbol{U} + \boldsymbol{A}\boldsymbol{U} = \mathbb{I} \qquad \boldsymbol{A}\boldsymbol{U} = \boldsymbol{0}$$
$$\boldsymbol{W}_{out}\boldsymbol{O}\boldsymbol{V} = \boldsymbol{U}^\dagger \boldsymbol{O}\boldsymbol{V} + \boldsymbol{A}\boldsymbol{O}\boldsymbol{V} = \boldsymbol{0} \qquad \boldsymbol{A}\boldsymbol{O}\boldsymbol{V} = -\boldsymbol{U}^\dagger \boldsymbol{O}\boldsymbol{V} \tag{55}$$

So $\boldsymbol{A}$ and $\boldsymbol{U}^\dagger$ live in orthogonal subspaces, and $\boldsymbol{A} \neq \boldsymbol{0}$. The same applies to $\boldsymbol{V}$ and $\boldsymbol{B}$, therefore:

$$||\boldsymbol{W}_{out}||_F^2 = ||\boldsymbol{U}^\dagger||_F^2 + ||\boldsymbol{V}^\dagger||_F^2 + ||\boldsymbol{A}||_F^2 + ||\boldsymbol{B}||_F^2 \geq ||\boldsymbol{U}^\dagger||_F^2 + ||\boldsymbol{V}^\dagger||_F^2 \tag{56}$$

And hence orthogonalising the subspaces reduces the loss in this case as well.

Combining these two, the optimal representation of range-independent multi-dimensional variables for activity losses using an Lp norm for $p > 1$ is modular.

## C    IMPERFECT RECONSTRUCTION

In our results we assume perfect reconstruction. This is not the real situation in either biology or neural networks. If a dataset is range-independent, but only when considering a very unlikely point, then in reality that point might be dropped, incurring a small reconstruction cost, in order to save firing and weight energy.

We illustrate this in the following example. Our dataset comprises four points from the corners of a square - the dataset is range independent. However, we sample these points very nonuniformly, one with probability $\Delta = 0.01$, the rest with probability $\Delta = \frac{1-\Delta}{3}$, Figure 7A. If the penalty on reconstruction error is high enough during numerical optimisation, all four points are perfectly reconstructed, this dataset is range independent, and the representation modularises Figure 7B. If the reconstruction loss is made less important the representation changes dramatically, and mixes Figure 7C.

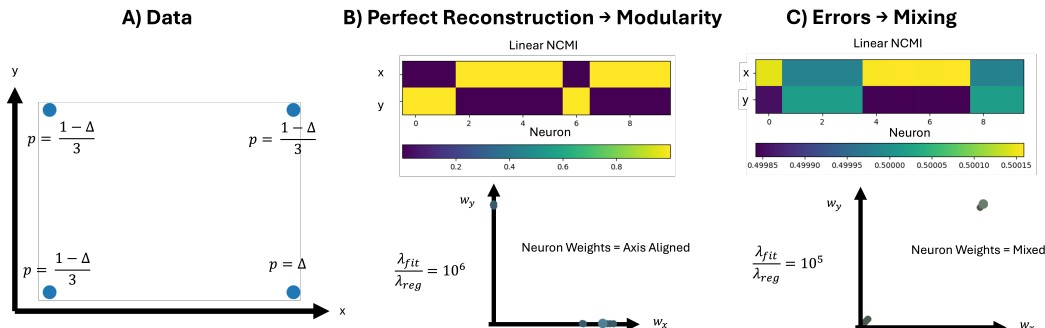

Figure 7: A) We sample a dataset of four points from the corners of a square, one with probability $\Delta$, the rest with probability $\frac{1-\Delta}{3}$. B) If the penalty on reconstruction error is high enough, the data fits perfectly, and the representation is modular. C) If the penalty is lowered, it is preferred to incur reconstruction cost, and mix the representation.

This is an interesting and very relevant effect, that future work could usefully characterise.

## D  Positive Linear Recurrent Neural Networks

One of the big advantages of changing the phrasing of the modularising constraints from statistical properties (as in Whittington et al. (2023)) to range properties is that it naturally generalises to recurrent dynamical formulations, and recurrent networks are often a much more natural setting for neuroscience. To illustrate this we'll study positive linear RNNs. Linear dynamical systems can only produce mixtures of decaying or growing sinusoids, so we therefore ask the RNN to produce different frequency outputs via an affine readout:

$$\boldsymbol{W}_{\text{out}}\boldsymbol{g}(t) + \boldsymbol{b}_{\text{out}} = \begin{bmatrix} \cos(\omega_1 t) \\ \cos(\omega_2 t) \end{bmatrix} \tag{57}$$

Then we'll assume the internal dynamical structure is a standard linear RNN:

$$\boldsymbol{W}_{\text{rec}}\boldsymbol{g}(t) + \boldsymbol{b} = \boldsymbol{g}(t + \Delta t) \tag{58}$$

We will study this two frequency setting and ask when the representation learns to modularise the two frequencies. These results could be generalised to multiple variables as in section B.5. Again, our representation must be non-negative, $\boldsymbol{g}(t) \geq \boldsymbol{0}$, and we minimise the following energy loss:

$$\mathcal{L} = \langle ||\boldsymbol{g}(t)||^2 \rangle + \lambda_W ||\boldsymbol{W}_{\text{rec}}||_F^2 + \lambda_R ||\boldsymbol{W}_{\text{out}}||_F^2 \tag{59}$$

We will show that, if $\lambda_W$ is sufficiently large and the RNN solves the task for infinite time, the optimal representation contains activity cycling at only the two frequencies, $\omega_1$ and $\omega_2$. Further we'll show that these two parts should modularise if the two frequencies are irrational ratios of one another, and should mix if one frequency is an integer multiple of the other, i.e. the frequencies are harmonics. Finally, we conjecture, show empirically, and present proof in limited cases, that non-harmonic rational frequency ratios should modularise. If the task runs for finite time then a smudge factor is allowed, the frequencies can be near integer multiples, and how close you have to be to mix is governed by the length of the task. As long as during the task the frequencies are 'effectively' integer multiples, the representation will mix.

To show this we will first study the weight losses, show that they are minimised if each frequency is modularised from the others. We will then argue that when $\lambda_W$ is sufficiently large the representation will contain only two frequencies. Finally, we will study when the activity loss is minimised. Then, as in Appendix B, when all losses agree on their minima, the representation modularises, else it mixes.

### D.1  Structure of Representation - Act I

Due to the autonomous linear system architecture, the representation can only contain a linear mixture of growing, decaying, or stable frequencies, and a constant. The growing and decaying modes cannot help you solve the task (since if they do at one point in time, they later won't), and will either cost infinite energy (the growing modes) or will cost some energy without helping to reduce the bias (since, again, the bias has to be set large enough such that after the decaying mode has disappeared the representation is still positive). Hence,

$$\boldsymbol{g}(t) = \sum_{i=1}^{F} \boldsymbol{a}_i \cos(\omega_i t) + \boldsymbol{b}_i \sin(\omega_i t) + \boldsymbol{b}_0 \tag{60}$$

Where $F$ is smaller than half the number of neurons to ensure the autonomous linear system can propagate each frequency (each frequency 'uses' 2 dimensions, as will be obvious from Appendix D.3.

### D.2  Readout Loss

The readout loss is relatively easy, again create some capitalised pseudoinverse vectors $\{\boldsymbol{A}_i, \boldsymbol{B}_i\}_{i=1}^{F}$ defined by being the min-norm vectors with the property that $\boldsymbol{A}_i^T \boldsymbol{a}_j = \delta_{ij}$ and $\boldsymbol{A}_i^T \boldsymbol{b}_j = 0$, and the same for $\boldsymbol{B}_i$. Then the min-norm readout matrix is:

$$\boldsymbol{W}_{\text{out}} = \begin{bmatrix} \boldsymbol{A}_1^T \\ \boldsymbol{A}_2^T \end{bmatrix} \tag{61}$$

And the readout loss is:

$$|\boldsymbol{W}_{\text{out}}|_F^2 = \text{Tr}[\boldsymbol{W}_{\text{out}}\boldsymbol{W}_{\text{out}}^T] = |\boldsymbol{A}_1|^2 + |\boldsymbol{A}_2|^2 \tag{62}$$

Each vector has the following form, $\boldsymbol{A}_i = \frac{1}{\boldsymbol{a}_i^2}\boldsymbol{a}_i + \boldsymbol{a}_{i,\perp}$, where $\boldsymbol{a}_{i,\perp}$ is orthogonal to $\boldsymbol{a}_i$ and is included to ensure the correct orthogonality properties hold. So, for a fixed encoding size, the readout loss is minimised if $\boldsymbol{a}_{i,\perp} = 0$. This occurs when the encodings are orthogonal (i.e. $\boldsymbol{a}_1$ and $\boldsymbol{a}_2$ are orthogonal from one another, all other $\boldsymbol{a}_i$ vectors, and from each of the $\boldsymbol{b}_i$ vectors). This happens if the two frequencies are modularised from one another, and additionally the sine and cosine vectors for each frequency are orthogonal. I.e. a modularised solution with this property has the minimal readout weight loss for a given encoding size.

### D.3   RECURRENT LOSS WITHOUT BIAS

First we consider the slightly easier case where there is no bias in the recurrent dynamics:

$$\boldsymbol{W}\boldsymbol{g}(t) = \boldsymbol{g}(t+1) \tag{63}$$

Then we will write down a convenient decomposition of the min-norm $\boldsymbol{W}$. Call the matrix of stacked coefficient vectors, $\boldsymbol{X}$:

$$\boldsymbol{X} = [\boldsymbol{a}_1 \quad \boldsymbol{b}_1 \quad \ldots \quad \boldsymbol{a}_F \quad \boldsymbol{b}_F \quad \boldsymbol{b}_0] \tag{64}$$

Similarly, call the matrix of stacked normalised psuedo-inverse vectors $\boldsymbol{X}^\dagger$:

$$\boldsymbol{X}^\dagger = [\boldsymbol{A}_1 \quad \boldsymbol{B}_1 \quad \ldots \quad \boldsymbol{A}_F \quad \boldsymbol{B}_F \quad \boldsymbol{B}_0] \tag{65}$$

And finally create an ideal rotation matrix:

$$\boldsymbol{R} = \begin{bmatrix} \cos(\omega_1\Delta t) & -\sin(\omega_1\Delta t) & \ldots & 0 & 0 & 0 \\ \sin(\omega_1\Delta t) & \cos(\omega_1\Delta t) & \ldots & 0 & 0 & 0 \\ \vdots & \vdots & \ddots & \vdots & \vdots & \vdots \\ 0 & 0 & \ldots & \cos(\omega_F\Delta t) & -\sin(\omega_F\Delta t) & 0 \\ 0 & 0 & \ldots & \sin(\omega_F\Delta t) & \cos(\omega_F\Delta t) & 0 \\ 0 & 0 & \ldots & 0 & 0 & 1 \end{bmatrix}$$
$$= \begin{bmatrix} \boldsymbol{R}_1 & 0 & \ldots & 0 & 0 \\ 0 & \boldsymbol{R}_2 & \ldots & 0 & 0 \\ \vdots & \vdots & \ddots & \vdots & \vdots \\ 0 & 0 & \ldots & \boldsymbol{R}_F & 0 \\ 0 & 0 & \ldots & 0 & 1 \end{bmatrix} \tag{66}$$

Where each $\boldsymbol{R}_i$ is a $2 \times 2$ rotation matrix at frequency $\omega_i$. Now:

$$\boldsymbol{W} = \boldsymbol{X}\boldsymbol{R}\boldsymbol{X}^{\dagger,T} \tag{67}$$

We can then calculate the recurrent weight loss:

$$|\boldsymbol{W}|_F^2 = \text{Tr}[\boldsymbol{W}\boldsymbol{W}^T] = \text{Tr}[\boldsymbol{X}^T\boldsymbol{X}\boldsymbol{R}\boldsymbol{X}^{\dagger,T}\boldsymbol{X}^\dagger\boldsymbol{R}^T] \tag{68}$$

$\boldsymbol{X}^T\boldsymbol{X}$ is a symmetric $2F + 1 \times 2F + 1$ positive-definite matrix, and its inverse is $\boldsymbol{X}^{\dagger,T}\boldsymbol{X}^\dagger$, another symmetric positive-definite matrix. To see that these matrices are inverses of one another perform the singular value decomposition, $\boldsymbol{X} = \boldsymbol{U}\Sigma\boldsymbol{V}^T$, and $\boldsymbol{X}^\dagger = \boldsymbol{U}\Sigma^{-1}\boldsymbol{V}^T$. Then $\boldsymbol{X}^T\boldsymbol{X} = \boldsymbol{V}\Sigma^2\boldsymbol{V}^T$ and $\boldsymbol{X}^{\dagger,T}\boldsymbol{X}^\dagger = \boldsymbol{V}\Sigma^{-2}\boldsymbol{V}^T$, which are clearly inverses of one another.

Introducing a new variable, $\boldsymbol{Y} = \boldsymbol{X}^T\boldsymbol{X}$:

$$|\boldsymbol{W}|_F^2 = \text{Tr}[\boldsymbol{Y}\boldsymbol{R}\boldsymbol{Y}^{-1}\boldsymbol{R}^T] \tag{69}$$

We then use the following trace inequality, from Ruhe (Ruhe, 1970). For two positive semi-definite symmetric matrices, $\boldsymbol{E}$ and $\boldsymbol{F}$, with ordered eigenvalues, $e_1 \geq \ldots \geq e_n \geq 0$ and $f_1 \geq \ldots \geq f_n \geq 0$

$$\text{Tr}[\boldsymbol{E}\boldsymbol{F}] \geq \sum_{i=1}^n e_i f_{n-i+1} \tag{70}$$

Now, since $\boldsymbol{R}\boldsymbol{Y}^{-1}\boldsymbol{R}^T$ and $\boldsymbol{Y}^{-1}$ are similar matrices, they have the same eigenvalues, and $\boldsymbol{Y}^{-1}$ is the inverse of $\boldsymbol{Y}$ so its eigenvalues are the inverse of those of $\boldsymbol{Y}$. Therefore:

$$|\boldsymbol{W}|_F^2 = \mathrm{Tr}[\boldsymbol{Y}\boldsymbol{R}\boldsymbol{Y}^{-1}\boldsymbol{R}^T] \geq \sum_{i=1}^{2F+1} \frac{\lambda_i}{\lambda_i} = 2F+1 \tag{71}$$

Then we can show that this lower bound on the weight loss is achieved when the coefficient vectors are orthogonal, hence making the modular solution optimal. If all the coefficient vectors are orthogonal then $\boldsymbol{Y}$ and $\boldsymbol{Y}^{-1}$ are diagonal, so they commute with any matrix, and:

$$|\boldsymbol{W}|_F^2 = \mathrm{Tr}[\boldsymbol{Y}\boldsymbol{R}\boldsymbol{Y}^{-1}\boldsymbol{R}^T] = \mathrm{Tr}[\boldsymbol{Y}\boldsymbol{Y}^{-1}\boldsymbol{R}^T\boldsymbol{R}] = \mathrm{Tr}[\mathbb{I}_{2F+1}] = 2F+1 \tag{72}$$

### D.4 Recurrent Loss with Bias

Now we return to the case of interest:

$$\boldsymbol{W}\boldsymbol{g}(t) + \boldsymbol{b} = \boldsymbol{g}(t+1) \tag{73}$$

Our energy loss, equation 59, penalises the size of the weight matrix $|\boldsymbol{W}|_F^2$ and not the bias. Therefore, if we can make $\boldsymbol{W}$ smaller by assigning some of its job to $\boldsymbol{b}$ then we should. We can do this by setting $\boldsymbol{b}_0 = \boldsymbol{b}$ (recall the definition of $\boldsymbol{b}_0$ from equation 60) and constructing the following, smaller, min-norm weight matrix:

$$\boldsymbol{W} = \begin{bmatrix} \boldsymbol{a}_1 & \boldsymbol{b}_1 & \dots & \boldsymbol{a}_F & \boldsymbol{b}_F \end{bmatrix} \begin{bmatrix} \boldsymbol{R}_1 & 0 & \dots & 0 \\ 0 & \boldsymbol{R}_2 & \dots & 0 \\ \vdots & \vdots & \ddots & \vdots \\ 0 & 0 & \dots & \boldsymbol{R}_F \end{bmatrix} \begin{bmatrix} \boldsymbol{A}_1^T \\ \boldsymbol{B}_1^T \\ \vdots \\ \boldsymbol{A}_F^T \\ \boldsymbol{B}_F^T \end{bmatrix} = \hat{\boldsymbol{X}}\hat{\boldsymbol{R}}\hat{\boldsymbol{X}}^\dagger \tag{74}$$

Using the definitions in the previous section. This slightly complicates our previous analysis because now $\hat{\boldsymbol{X}}^{\dagger,T}\hat{\boldsymbol{X}} = \hat{Y}^\dagger$ is not the inverse of $\hat{\boldsymbol{X}}^T\hat{\boldsymbol{X}} = \hat{Y}$. If $\boldsymbol{b}$ is orthogonal to all the vectors $\{\boldsymbol{a}_i, \boldsymbol{b}_i\}_{i=1}^F$, then it is the inverse, and the previous proof that modularity is an optima goes through.

Fortunately, it is easy to generalise to this setting. $\boldsymbol{X}^\dagger$ is not quite the pseudoinverse of $\boldsymbol{X}^\dagger$ because its vectors have to additionally be orthogonal to $\boldsymbol{b}$. This means we can break down each of the vectors into two components, for example:

$$\boldsymbol{A}_1 = \hat{\boldsymbol{A}}_1 + \hat{\boldsymbol{A}}_{1,\perp} \tag{75}$$

The first of these vectors is the transpose of the equivalent row of the pseudoinverse of $\hat{\boldsymbol{X}}$, it lives in the span of the vectors $\{\boldsymbol{a}_i, \boldsymbol{b}_i\}_{i=1}^F$. The second component is orthogonal to this span and ensures that $\boldsymbol{A}_1^T\boldsymbol{b} = 0$. Hence, the previous claim. If $\boldsymbol{b}$ is orthogonal to the span of $\{\boldsymbol{a}_i, \boldsymbol{b}_i\}_{i=1}^F$, this is the standard pseudoinverse and the previous result goes through.

We can express the entire $\hat{\boldsymbol{X}}^\dagger$ matrix in these two components:

$$\hat{\boldsymbol{X}}^\dagger = \hat{\boldsymbol{X}}_0^\dagger + \hat{\boldsymbol{X}}_\perp^\dagger \tag{76}$$

Then:

$$\hat{\boldsymbol{Y}}^\dagger = (\hat{\boldsymbol{X}}_0^\dagger + \hat{\boldsymbol{X}}_\perp^\dagger)^T(\hat{\boldsymbol{X}}_0^\dagger + \hat{\boldsymbol{X}}_\perp^\dagger) = \hat{\boldsymbol{X}}_0^{\dagger,T}\hat{\boldsymbol{X}}_0^\dagger + \hat{\boldsymbol{X}}_\perp^{\dagger,T}\hat{\boldsymbol{X}}_\perp^\dagger = \hat{\boldsymbol{Y}}^{-1} + \hat{\boldsymbol{Y}}_\perp^\dagger \tag{77}$$

Both of these new matrices are positive semi-definite matrices, since they are formed by taking the dot product of a set of vectors. Then:

$$|\boldsymbol{W}|_F^2 = \mathrm{Tr}[\hat{\boldsymbol{Y}}\hat{\boldsymbol{R}}\hat{\boldsymbol{Y}}^{-1}\hat{\boldsymbol{R}}^T] + [\hat{\boldsymbol{Y}}\hat{\boldsymbol{R}}\hat{\boldsymbol{Y}}_\perp^\dagger\hat{\boldsymbol{R}}^T] \tag{78}$$

Now, the first term is greater than or equal than $2F$, as in the previous setting. And since $\hat{\boldsymbol{Y}}$ and $\hat{\boldsymbol{Y}}_\perp^\dagger$ are positive semi-definite, and so therefore is $\hat{\boldsymbol{R}}\hat{\boldsymbol{Y}}_\perp^\dagger\hat{\boldsymbol{R}}^T$, this second term is greater than or equal to 0. Hence, $|\boldsymbol{W}|_F^2 \geq 2F$, and orthogonal encodings achieve this bound, therefore it is an optimal solution according to the weight loss.

### D.5 STRUCTURE OF REPRESENTATION - ACT II - LARGE $\lambda_W$

We saw that our representation takes the following form.

$$\boldsymbol{g}(t) = \sum_{i=1}^{F} \boldsymbol{a}_i \cos(\omega_i t) + \boldsymbol{b}_i \sin(\omega_i t) + \boldsymbol{b}_0 \tag{79}$$

Further, and we saw in the previous section that each added frequency increases the recurrent weight loss. To solve the task we only need two frequencies, why would we have more than two? The only possibility is that by including additional frequencies we might save activity energy, at the cost of weight energy. We will simplify our life for now by saying $\lambda_W$ is very large, so our representation only has two frequencies. We now turn to our activity loss and ask which frequencies should mix, and which modularise.

### D.6 FREQUENCIES

Consider a mixed frequency neuron:

$$g_i(t) = (\boldsymbol{a}_1)_i \cos(\omega_1 t) + (\boldsymbol{b}_1)_i \sin(\omega_1 t) + (\boldsymbol{a}_2)_i \cos(\omega_2 t) + (\boldsymbol{b}_2)_i \sin(\omega_2 t) + b_n \tag{80}$$

By rescaling and shifting time, we can rewrite all such mixed neurons in a simpler form:

$$g_i(t) = \alpha_i \cos(t) + \beta_i \cos(\omega t + \phi) + b_i \quad \omega > 1 \tag{81}$$

Where $\omega$ is the ratio of the larger to the smaller frequencies. We will show that the activity of this mixed neuron is lower than its corresponding modular counterpart when $\omega$ is an integer, and higher if $\omega$ is irrational. Finally, we show empirically, and some theoretical evidence, that the same is true of rational, non-integer $\omega$. Hence, we find the activity energy is minimised by modularising unless one of the frequencies is an integer multiple of the others. Further, since if the mixed neuron $g_i(t)$ is preferred over its modular counterpart, then so too are positively scaled versions, $\mu g_i(t)$, we will consider responses of the form:

$$g(t) = \cos(t) + \delta \cos(\omega t + \phi) + \Delta \quad \omega > 1 \tag{82}$$

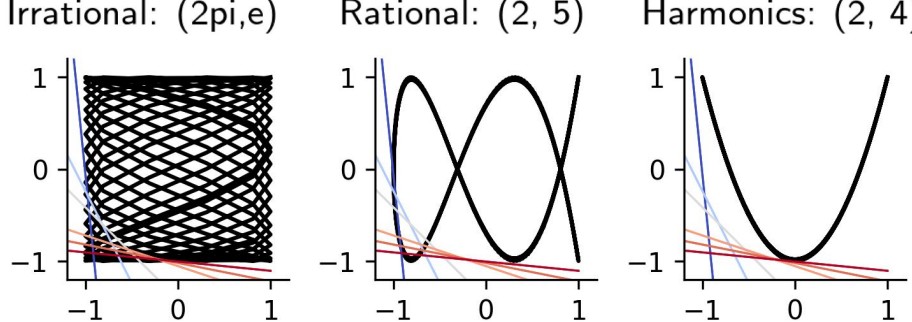

Figure 8: Schematics showing irrational and rational ratio case the data are range dependent beyond the modular-mixed boundary while in de-modularising harmonics case, the two periodic waves are range independent.

**Irrational Frequencies Modularise** If $\omega$ is irrational then, by Kronecker's theorem, you can find a value of $t$ for which $\cos(t)$ and $\cos(\omega t + \phi)$ take any pair of values. This makes the two frequencies, among other things, extreme point independent, and therefore no mixing is better than modularising.

**Even Integer Multiples Mix** According to Theorem 2.1, to modularise, for all $\delta$ and $\phi$ values:

$$\Delta > \sqrt{1 + \delta^2} = 1 + \mathcal{O}(\delta^2) \tag{83}$$

We will therefore try to find a $\delta$ and $\phi$ which breaks this inequality. Consider $\delta < 0$ and very small and $\phi = 0$, then, writing $\omega = 2m$ for integer $m$:

$$\Delta = -\min_i [\cos(t) + \delta \cos(2mt)] \tag{84}$$

$\cos(t) - |\delta| \cos(2mt)$ takes its extreme values when $t$ is an integer multiple of $\pi$, and the smallest it can be is $-1 + |\delta|$. Hence:

$$\Delta = 1 - \delta < 1 + \mathscr{O}(\delta^2) \tag{85}$$

This is smaller than the critical value, so the representation should mix, since at least one mixing inequality was broken.

**Odd Integer Multiples Mix**    If $\omega$ is an odd integer then we can instead mix cosine with sine by choosing $\phi = \frac{\pi}{2}$:

$$g_i(t) = \cos(t) + \delta \sin(\omega t) + \Delta \tag{86}$$

Then the same argument goes through as above.

**Other Rational Multiples Modularise**    Now consider $\omega = \frac{p}{q}$ for two integers $p$ and $q \neq 1$. We were inspired in this section by the mathoverflow post of Soudry and Speiser (2015). So we are considering the mixed encoding:

$$g(t) = \cos(t) + \delta \cos(\frac{p}{q}t + \phi) + \Delta \tag{87}$$

And we have to show that for all $\delta$ and $\phi$, $\Delta^2 > 1 + \delta^2$. For general $\phi$ we have been unable to do this, but for $\phi = 0$ we present proof below in the hope this can be generalised.

We break down the problem into four cases based on the sign of $\delta$, and whether $p$ and $q$ are both odd, or only one is. The easiest is if both are odd, and $\delta > 0$. Then take $t = q\pi$:

$$g(t) = \cos(q\pi) + \delta \cos(p\pi) + \Delta = -1 - \delta + \Delta > 0 \tag{88}$$

Therefore $\Delta$ is at least $1 + \delta$, which is larger than $\sqrt{1 + \delta^2}$, hence modularising is better.

Conversely, if one of $p$ or $q$ is even (let's say $p$ w.l.o.g.) and $\delta < 0$ then choose $t = p\pi$ again:

$$g(t) = -1 + \delta \cos(p\pi) + \Delta = -1 - |\delta| + \Delta > 0 \tag{89}$$

And the same argument holds.

Now consider the case where $\delta > 0$ and one of $p$ or $q$ is odd. Then there exists an odd integer $k$ such that Soudry and Speiser (2015):

$$kp = q + 1 \mod 2q \tag{90}$$

Therefore take $t = k\pi$:

$$g(t) = \cos(k\pi) + \delta \cos(\pi(\frac{kp}{q})) + \Delta = -1 + \delta \cos(\pi(\frac{q+1+2nq}{q})) + \Delta \tag{91}$$

For some integer $n$. Developing:

$$g(t) = -1 - \delta \cos(\frac{\pi}{q}) + \Delta \tag{92}$$

The key question is whether for any $\Delta$ below the critical value this representation is nonnegative. For this to be true:

$$-1 - \delta \cos(\frac{\pi}{q}) + \sqrt{1 + \delta^2} \geq 0 \tag{93}$$

Developing this we get a condition on the mixing coefficient $\delta$:

$$\delta \geq \frac{2\cos(\frac{\pi}{q})}{1 - \cos^2(\frac{\pi}{q})}. \tag{94}$$

We can apply a similar argument with $t = \frac{qk\pi}{p}$ to get:

$$-\cos(\frac{\pi}{p}) - \delta + \Delta \geq 0 \tag{95}$$

This leads us to:

$$\delta \leq \frac{1 - \cos^2(\frac{\pi}{p})}{2\cos(\frac{\pi}{p})} \tag{96}$$

We squeeze inequalities eqn.94,96 and get

$$1 \leq \frac{(1 - \cos^2(\pi/p))(1 - \cos^2(\pi/q))}{4\cos(\pi/p)\cos(\pi/q)} \tag{97}$$

For any even, odd integer pair of $p, q \neq 1$ and $\delta > 0$, $\cos(\pi/p)$ and $\cos(\pi/q)$ are bounded in $(0, 1]$. In this case, the maximum possible value of the righthand side is achieved with the smallest possible pair of integers $(3, 4)$, for which the inequality does not hold. Thus the inequalities eqn.96, 94 cannot hold at the same time which let us to conclude that there is no $\Delta$ below the critical value satisfying nonnegativity constraint and leads to modular representation.

The last case is where $\delta < 0$ and both $p, q$ are odd. We can extend the relation shown in eqn.90 as following by simple substitution of oddity of the variables. With given odd integers $p, q$, there exists an *even* integer $k$ satisfying eqn.90. Similar to above, take $t = \frac{qk\pi}{p}$ for the mixed encoding eqn.87 and we develop:

$$g_i = -\cos(\frac{\pi}{p}) + \delta + \Delta \tag{98}$$

which leads to the following inequality:

$$\delta \geq \frac{1 - \cos^2(\pi/p)}{-2\cos(\pi/p)} \tag{99}$$

Now take $t = k\pi$, and we get:

$$g_i = 1 - \delta\cos(\pi/q) + \Delta \tag{100}$$

which leads to the in equality,

$$\delta \leq \frac{-2\cos(\pi/q)}{1 - \cos^2(\pi/q)} \tag{101}$$

Again, the above two inequalities cannot hold together with the odd integers $p, q > 1$.

We also empirically show on Fig 8 that in rational multiple frequencies, there is no modular-mixed boundary possible to break the modularisation.

# E    METRICS FOR REPRESENTATIONAL MODULARITY AND SOURCE STATISTICAL INTERDEPENDENCE

We seek to design a metric to measure how much information a latent contains about a source. However, we would like to ignore information that could be explained through the latent's tuning to a different source. For example, perhaps two variables are correlated, if a latent encodes one of them, then it will also be slightly informative about the other despite not being functionally related. We therefore seek to condition on all of the sources bar one, and measure how much information the latent contains about the remaining source.

Dunion et al. (2023) leverage conditional mutual information in a similar way to this in a reinforcement learning context, but for training rather than evaluation, and therefore resort to a naive Monte Carlo estimation scheme that scales poorly. Instead, we leverage the identity

$$I(\boldsymbol{z}_j; \boldsymbol{s}_i | \boldsymbol{s}_{-i}) = I(\boldsymbol{z}_j; \boldsymbol{s}) - I(\boldsymbol{z}_j; \boldsymbol{s}_{-i}), \tag{102}$$

where $\boldsymbol{s}_{-i}$ is a shorthand for $\{\boldsymbol{s}_{i'} \mid i' \neq i\}$. Since this involves computing mutual information with multiple sources, we restrict ourselves to considering discrete sources and use a continuous-discrete KSG scheme (Ross, 2014) to estimate information with continuous neural activities. We normalise conditional mutual information by $H(\boldsymbol{s}_i | \boldsymbol{s}_{-i})$ to obtain a measure in $[0, 1]$. We then arrange the pairwise quantities into a matrix $\boldsymbol{C} \in \mathbb{R}^{d_s \times d_z}$ and compute the normalised average "max-over-sum" in a column as a measure of sparsity, following Hsu et al. (2023):

$$\text{CInfoM}(\boldsymbol{s}, \boldsymbol{z}) := \left( \frac{1}{d_z} \sum_{j=1}^{d_z} \frac{\max_i \boldsymbol{C}_{ij}}{\sum_{i=1}^{d_s} \boldsymbol{C}_{ij}} - \frac{1}{d_s} \right) \bigg/ \left( 1 - \frac{1}{d_s} \right). \tag{103}$$

CInfoM is appropriate for detecting arbitrary functional relationships between a source and a neuron's activity. However, in some of our experiments, the sources are provided as supervision for a linear readout of the representation. In such cases, the network cannot use information that is nonlinearly encoded, so the appropriate meausure is the degree of linearly encoded information. Operationalising this we leverage the predictive $\mathscr{V}$-information framework of Xu et al. (2020). We specify the function class $\mathscr{V}$ as linear and calculate

$$\begin{aligned} I_{\mathscr{V}}(\boldsymbol{z}_j \to \boldsymbol{s}_i | \boldsymbol{s}_{-i}) &= I_{\mathscr{V}}(\boldsymbol{z}_j \to \boldsymbol{s}) - I_{\mathscr{V}}(\boldsymbol{z}_j \to \boldsymbol{s}_{-i}) \\ &= H_{\mathscr{V}}(\boldsymbol{s}) - H_{\mathscr{V}}(\boldsymbol{s}|\boldsymbol{z}_j) - H_{\mathscr{V}}(\boldsymbol{s}_{-i}) + H_{\mathscr{V}}(\boldsymbol{s}_{-i}|\boldsymbol{z}_j), \end{aligned} \tag{104}$$

followed by a normalisation by $H_{\mathscr{V}}(\boldsymbol{s}_i | \boldsymbol{s}_{-i})$. Each predictive conditional $\mathscr{V}$-entropy term is estimated via a standard maximum log-likelihood optimization over $\mathscr{V}$, which for us amounts to either logistic regression or linear regression, depending on the treatment of the source variables as discrete or continuous. The pairwise linear predictive conditional information quantities are reduced to a single linear conditional InfoM quantity by direct analogy to (103).

Finally, to facilitate comparisons across different source distributions, we report CInfoM against the normalised multiinformation of the sources:

$$\text{NI}(\boldsymbol{s}) = \frac{\sum_{i=1}^{d_s} H(s_i) - H(\boldsymbol{s})}{\sum_{i=1}^{d_s} H(s_i) - \max_i H(s_i)}. \tag{105}$$

This allows us to test the following null hypothesis: that breaking statistical independence, rather than range independence, is more predictive of mixing. On the other hand, if range independence is more important, then source distributions that retain range independence while admitting nonzero multiinformation will better induce modularity compared to those that break range independence.

# F WHAT-WHERE TASK

## F.1 EXPERIMENTAL SETUP

### F.1.1 DATA GENERATION

The network modularises when exposed to both simple and complex shapes. Simple shapes, used in all main paper figures, are 9-element one-hots, reflecting the active pixel's position in a $3 \times 3$ grid. Formally, for a shape at position $(i, j)$ in the grid, the corresponding vector $s \in \mathbb{R}^9$ is given by:

$$s_k = \begin{cases} 1 & \text{if } k = 3(i-1) + j \\ 0 & \text{otherwise} \end{cases}$$

where $i, j \in \{1, 2, 3\}$.

For complex shapes, each shape is a binary vector $c \in \mathbb{R}^9$, with exactly 5 elements set to 1 and 4 elements set to 0, representing the active and inactive pixels, respectively. Each complex shape takes the (approximate) shape of a letter (9) and can be shifted to any of the 9 positions in the $3 \times 3$ grid. The standard training and testing datasets include one of every shape-position pair. The correlations introduced later include duplicates of a subset of these data points.

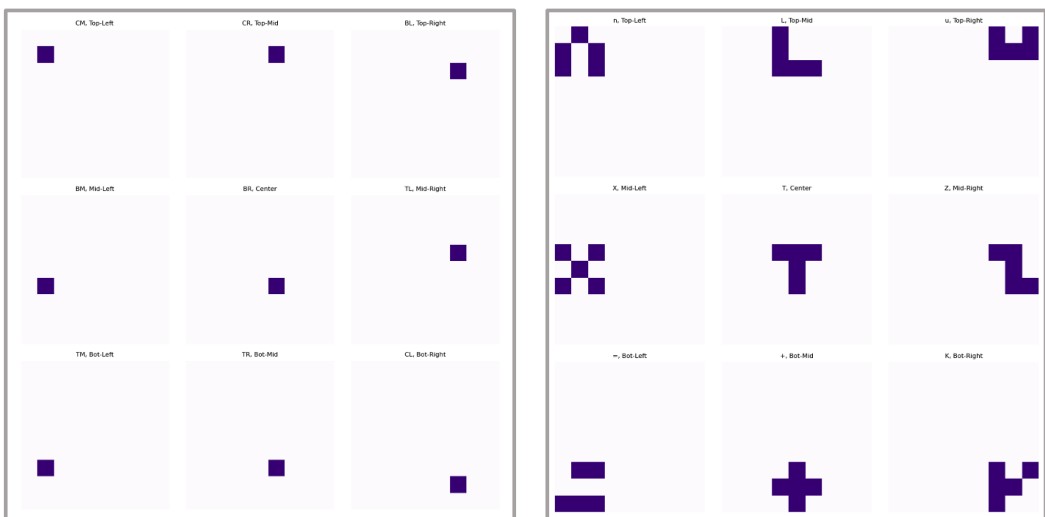

Figure 9: Different possible inputs to the network. Both one-hot (left) and letter-based (right) shapes can be outputted as either a concatenation of one-hots or as a 2D variable.

### F.1.2 NETWORK ARCHITECTURE

The network we use here is designed in PyTorch and takes as input am 81-element vector, flattened to a $9 \times 9$ image (9).The network architecture is formally defined as follows:

$$\text{Input:} \quad x \in \mathbb{R}^{81}$$
$$\text{Hidden layer:} \quad \mathbf{h} = \phi(\mathbf{W}_1 x + \mathbf{b}_1), \quad \mathbf{W}_1 \in \mathbb{R}^{25 \times 81}, \quad \mathbf{b}_1 \in \mathbb{R}^{25}$$
$$\text{Output layer:} \quad \mathbf{y} = \mathbf{W}_2 h + \mathbf{b}_2, \quad \mathbf{W}_2 \in \mathbb{R}^{2 \times 25}, \quad \mathbf{b}_2 \in \mathbb{R}^2$$

where $\phi$ is the activation function, either ReLU or $\tanh$, depending on the experiment. Weights $\mathbf{W}_1$ and $\mathbf{W}_2$ are initialised with a normal distribution $\mathcal{N}(0, 0.01)$, and biases $\mathbf{b}_1$ and $\mathbf{b}_2$ are initialised to zero.

### F.1.3 TRAINING PROTOCOLS

The network uses the Adam optimiser (Kingma and Ba (2014)), with learning rate and other hyper-parameters varying from experiment to experiment. The mean squared error (MSE) is calculated

separately for 'what' and 'where' tasks, and then combined along with the regularisation terms. The total loss function $L_{total}$ is a combination of the task-specific losses and regularisation terms:

$$L_{total} = L_{what} + L_{where} + \lambda_R(||\mathbf{W}_1||_2^2 + ||\mathbf{W}_2||_2^2 + ||\mathbf{h}||_2^2)$$

where $L_{task}$ for 'what' and 'where' tasks is defined as:

$$L_{task}(y, \hat{y}) = \frac{1}{N} \sum_{i=1}^{N} (y_i - \hat{y}_i)^2$$

Where $\lambda_R$ is the regularisation hyperparameter, typically set to $0.01$ unless specified otherwise. The network is trained using the Adam optimiser with a learning rate ranging between $0.001$ and $0.01$, adjusted as needed. Experiments are run for order $10^4$ epochs on 5 random seeds. Each experiment was executed using a single consumer PC with 8GB of RAM and across all settings, the networks achieve negligible loss.

## F.2  MODULARITY OF WHAT & WHERE

### F.2.1  BIOLOGICAL CONSTRAINTS ARE NECESSARY FOR MODULARITY

To understand the effect of biological constraints on modularisation, consider the optimisation problem under positivity and energy efficiency constraints. The activation function $\phi(x)$ ensures non-negativity:

$$\phi(x) = \max(0, x)$$

The energy efficiency is enforced by adding an L2 regularisation term to the loss function. By minimising the combined loss function $L_{total}$, the network encourages sparse and low-energy activations, leading to a separation of neurons responding to different tasks ('what' and 'where').

To illustrate the necessity of biological constraints in the modularisation of these networks, we show below the weights and activity responses of networks for networks where these constraints aren't present. As discussed above, positivity is introduced via the ReLU activation function, and energy efficiency is defined as an L2 regularisation on the hidden weights and activities.

As shown in the neural tuning curves, the unconstrained networks do encode task features, but they do so such that each neuron responds to the specific values of both input features, they are mixed selective. Compare these results to the weights and activity responses when positivity and energy efficiency constraints are introduced. In this setting, there is a clear separation of 'what' and 'where' tasks, into two distinct sub-populations of neurons.

## F.3  DROPOUT & CORRELATION

We use two different approaches to dropout in order to illustrate the importance of range independence.

**Diagonal Dropout**: The first approach removes example data points (i.e., shape-position pairs) from elements along the diagonals, starting in the middle and extending out to all four corners. This has the effect of increasing the mutual information between data sources but does not significantly affect their range dependence. Formally, let $D$ be the set of all shape-position pairs. In the diagonal dropout setting, we remove pairs $(s_i, p_i)$ where $i$ lies along the diagonal of the input grid:

$$D' = D \setminus \{(s_i, p_i) \mid i \in \text{diagonal positions}\}$$

In the most extreme case, only one data point from each corner is removed, and this is not sufficient to force mixed-selectivity in the neurons.

**Corner Dropout**: The second approach removes data points from one corner of the distribution. This also increases the mutual information between sources but changes their extreme-point dependence. Specifically, we remove pairs $(s_i, p_i)$ where $i$ lies in the bottom-left corner of the input grid:

$$D'' = D \setminus \{(s_i, p_i) \mid i \in \text{bottom-left corner positions}\}$$

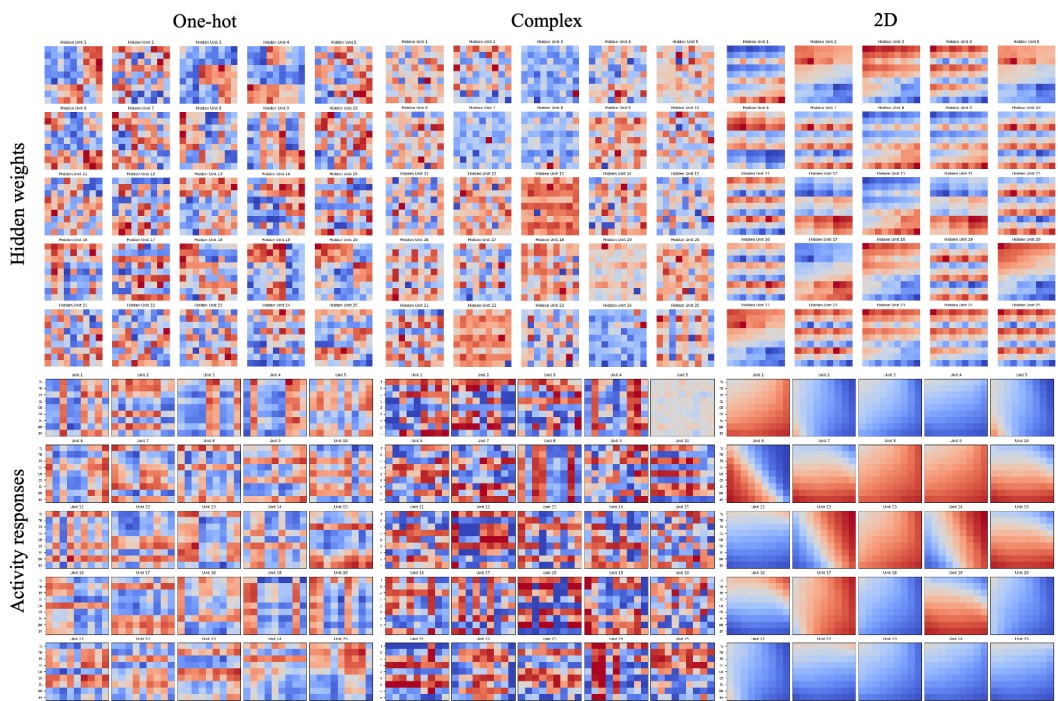

Figure 10: Hidden weights and activity responses of FF networks without biologically-inspired constraints. The one-hot and letter-like binary cases are shown, as well the 2D output setting with one-hot inputs.

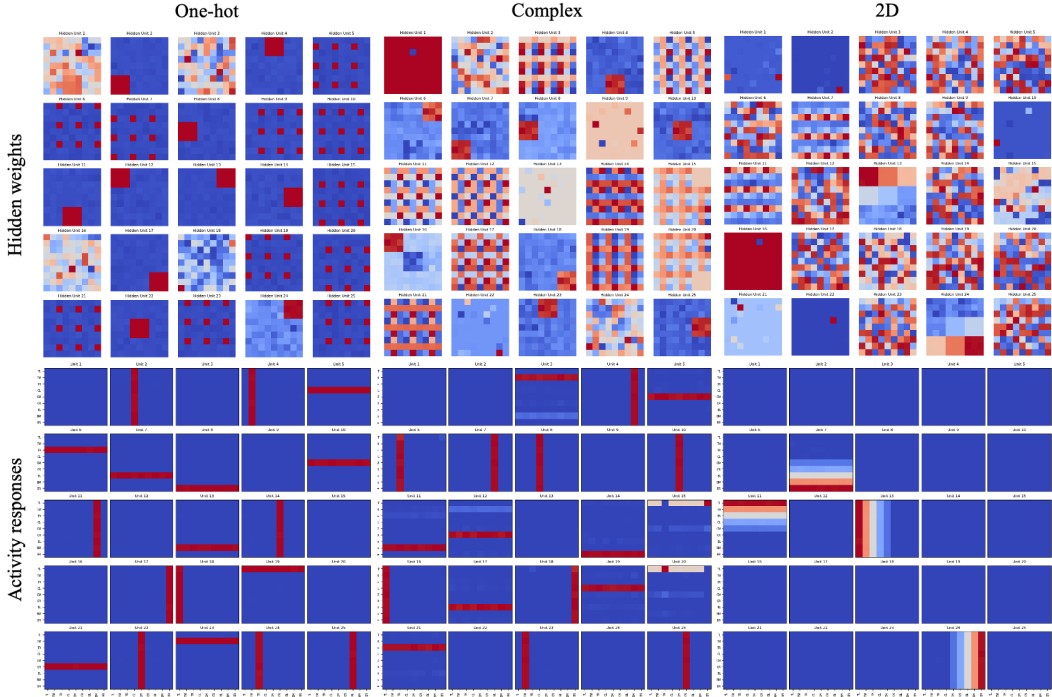

Figure 11: Weights and activity response of networks under biological constraints. The one-hot and letter-like binary cases are shown, as well the 2D output setting with one-hot inputs.

In this setting, a single data point removed from the corner is insufficient for breaking modularity; however, removing more than this causes mixed selective neurons to appear.

**Correlation**: To correlate sources, we duplicate data points that appear along the diagonal. This increases the mutual information between sources without affecting their range independence:

$$D''' = D \cup \{(s_i, p_i) \mid i \in \text{diagonal positions}\}$$

The mutual information $I(X; Y)$ between the shape $X$ and position $Y$ is calculated as follows:

$$I(X; Y) = \sum_{x,y} P(x, y) \log \frac{P(x, y)}{P(x)P(y)}$$

where $P(x, y)$ is the joint probability distribution of $X$ and $Y$.

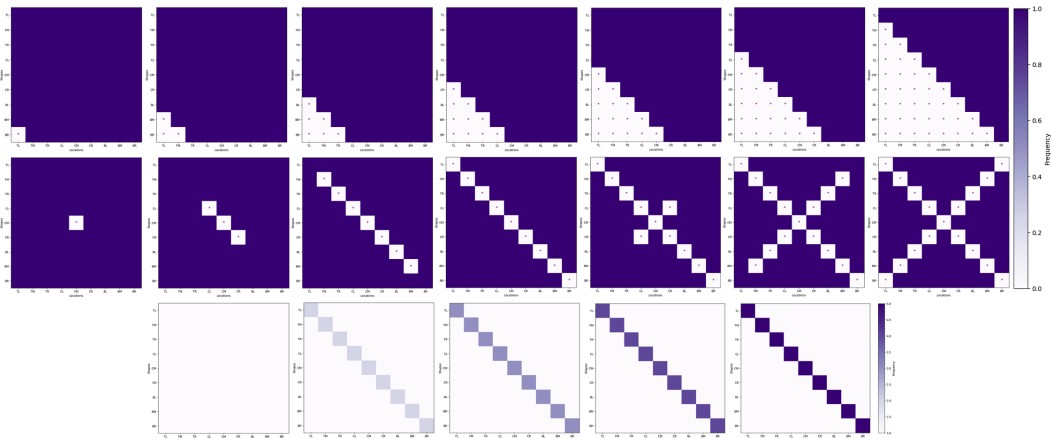

Figure 12: Increasing dropout (left to right) for both the corner-cutting (top) and diagonal (middle) cases, as well as correlation distributions (bottom). Note: asterisk denotes the absence of training data for this pair of input features.

## G  NONLINEAR AUTOENCODERS

### G.1  AUTOENCODING SOURCES

Three-dimensional source data is sampled from $[0, 1]^3$ and discretised to 21 values per dimension. The encoder and decoder are each a two-layer MLP with hidden size 16 and ReLU activation. The latent bottleneck has dimensionality six. All models use $\lambda_{\text{reconstruct}} = 1$, $\lambda_{\text{activity energy}} = 0.01$, $\lambda_{\text{activity nonnegativity}} = 1$, and $\lambda_{\text{weight energy}} = 0.0001$. Models are initialised from a He initialisation scaled by 0.3 and optimized with Adam using learning rate 0.001. Each experiment was executed using a single consumer GPU on a HPC with 2 CPUs and 4GB of RAM.

### G.2  AUTOENCODING IMAGES

We subsample 6 out of the 9 sources in the Isaac3D dataset, fixing a single value for the other 3. This yields a dataset of size $12,288$. We use an expressive convolutional encoder and decoder taken from the generative modeling literature. We use 12 latents, each quantised to take on 10 possible values. We use a weight decay of 0.1 and a learning rate of 0.0002 with the AdamW optimizer. Each experiment was executed using a single consumer GPU on a HPC with 8 CPUs and 8GB of RAM.

# H  RECURRENT NEURAL NETWORKS

## H.1  LINEAR AND NONLINEAR PERIODIC WAVE RNN EXPERIMENT DETAILS

For the linear RNN setting we provided a periodic pulse input (2D delta function of frequencies $w_1, w_2$ as input $x$, i.e. $x_k(t) = \delta(\cos(w_k t) - 1), k \in \{1, 2\}$) and trained the network to generate a two cosines of the same frequencies, $y_k(t) = \cos(w_k t)$.

For the nonlinear RNN we designed a two frequency mixing task. Given two source frequencies, the network must generate two cosine waves whose frequencies are the sum and difference of two input frequencies. The network requires non-linearities in order to approximate the multiplication: $\cos(a + b) = \cos(a)\cos(b) - \sin(a)\sin(b), \cos(a - b) = \cos(a)\cos(b) + \sin(a)\sin(b)$. In the task we provide the RNN with 2D periodic delta pulse of frequency $\frac{w1+w2}{2}$ and $\frac{w1-w2}{2}$ and learns to generate a trajectory of $\cos(w_1)$ and $\cos(w_2)$.

We use the following recurrent neural network,

$$\mathbf{g}(t+1) = f(\mathbf{W}_{\text{rec}}\mathbf{g}(t) + \mathbf{W}_{\text{in}} + \mathbf{b}_{\text{rec}}) \tag{106}$$

$$\mathbf{y}(t) = \mathbf{R}\mathbf{g}(t) + \mathbf{b}_{\text{out}} \tag{107}$$

in our linear RNNs $f(\cdot)$ is identity whereas in the nonlinear ones we used ReLU activation to enforce positivity condition.

For irrational output frequency ratio case, we used

$$w_1 = p\pi, w_2 = \sqrt{q}, p \sim \mathscr{U}(0.5, 4), q \sim \mathscr{U}(1, 10). \tag{108}$$

For rational case, we sampled

$$w_1, w_2 \in [1, 20] \cap \mathbb{Z}. \tag{109}$$

Where $\mathbb{Z}$ is the set of integers. For harmonics:

$$w_1 \in [1, 10] \cap \mathbb{Z}, w_2 = 2w_1. \tag{110}$$

We trained the RNN with the trajectory of length $T = 200$, bin size 0.1, and hidden dimension 16. For the linear RNN, we used learning rate 1e-3, 30k training iterations and $\lambda_{\text{target}} = 1, \lambda_{\text{activity}} = 0.5, \lambda_{\text{positivity}} = 5$, and $\lambda_{\text{weight}} = 0.02$. For the nonlinear RNN, we used learning rate 7.5e-4, 40k training iterations and $\lambda_{\text{target}} = 5, \lambda_{\text{activity}} = 0.5$ and $\lambda_{\text{weight}} = 0.01$. In both case, we initialised the weights to be orthogonal and the biases at zero, and used the Adam optimiser.

To assess the modularity of the trained RNNs, we performed a Fast Fourier Transform(FFT) on each neuron's activity and measured the relative power of the key frequencies $w_1, w_2$ with respect to the sum of total power spectrum

$$C_{\text{neuron}_i, w_j} = \frac{|FFT(g_i; w_j)|}{\sum_f |FFT(g_i; f)|} \tag{111}$$

and used it as a proxy of mutual information for modularity metric introduced in eqn. 103.

## H.2  NONLINEAR TEACHER-STUDENT RNNS

**Network details.** The Teacher network has input, hidden, and output dimensions of 2, and has orthogonal recurrent weights, input weights, and output weights. The Student RNN has input dimension 2, output dimension 2, and hidden dimension 64. It is initialised as per PyTorch default settings. The Teacher network dynamics is a vanilla RNN: $\boldsymbol{h}_t = tanh(\boldsymbol{W}_{rec}\boldsymbol{h}_{t-1} + \boldsymbol{W}_{in}\boldsymbol{i}_t)$, and each teacher predicts a target via $\boldsymbol{o}_t = \boldsymbol{W}_{out}\boldsymbol{h}_t$. The Student RNN has identical dynamics (but with a ReLU activation function, and different weight matrices etc).

**Generating training data.** The teacher RNN generates training data for the Student RNN. We want to tightly control the Teacher RNN hidden distribution (for corner cutting or correlation analyses), i.e., tightly control the source distribution for the training data. To control the distribution of hidden activities of the Teacher RNN, we use the following procedure. 1) We sample a randomly initialised Teacher RNN. 2) $\boldsymbol{h}_0$ is initialised as a vector of zeros. 3) With a batch size of $N$, we take a **single** step of the Teacher RNN (starting from 0 hidden state) assuming $\boldsymbol{i}_t = 0$. This produces network

activations (for each batch), $p_t = tanh(W_{rec}h_{t-1}$. 4) We then sample from idealised distribution of the teacher hidden states, $h_t$. For example a uniform distribution, or a cornet cut distribution. At this point these are just i.i.d. random variables, and not recurrently connected. 5) To recurrently connect these points, we optimise the input to the RNN, $i_t$, such that the RNN prediction, $p_t$, becomes, $h_t$. We then repeat steps 3-5) for all subsequent time-steps, i.e., we find what the appropriate inputs are to produce hidden states as if they were sampled from an idealised distribution. To prevent the Teacher RNN from being input driven, on step 4), we solve a linear sum assignment problem across all batches ($N$ batches), so the RNN (on average) gets connected to a sample, $h_t$, that is close to its initial prediction, $p_t$. This means the input, $i_t$, will be as small as possible and thus the Teacher RNN dynamics are as unconstrained as possible.

**Training.** We train the Student RNN on 10000 sequences generated by the Teacher RNN. The learning objective is a prediction loss $|o_t^{teacher} - o_t^{student}|^2$ plus regularisation of the squared activity of each neuron as well as each synapse (both regularisation values of 0.1). We train for 60000 gradient updates, with a batch size of 128. We use the Adam optimiser with learning rate 0.002.

### H.3   RNN MODELS OF ENTORHINAL CORTEX

We now talk through the linear RNNs used in Section 5. Our model is inspired by the entorhinal literature and linear network models of it Dorrell et al. (2023); Whittington et al. (2020; 2025; 2023). In each trial the agent navigates a 3x3 periodic environment. The RNN receives one special input only at timepoint 0 that tells it the layout of the room (format described later). Otherwise at each timestep it is told which action (north, south, east, west), the agent took. It uses this action to linearly update its hidden state via an action-dependent affine transform:

$$g_t = W_{a_t}g_{t-1} + b_{a_t} \tag{112}$$

At each timestep it has to output, via an affine readout, a particular target depending on the task.

In the spatial-reward task, Figure 4, at timepoint 0 the agent starts at a consistent position and receives an input that is a 9-dimensional 1-hot code telling the agent the relative position of the reward. At each timestep the agent then has to output two 9-dimensional 1-hot codes, one signalling its position in the room, the other the relative displacement of the reward from its position.

The agent experiences many rooms. Across rooms the rewards are either randomly sampled (random rooms), or sampled from one of two fixed positions (fixed rooms). Each agent experiences a different mixture of randomly sampled or fixed rooms. We then optimise the network to perform the task with non-negative neural activities while penalising the L2 norm of the activity and all weight matrices.

In the mixed-selectivity task, Figure 5, there are three objects in each room and the agent has to report the relative displacement of each of them. The input at the start of each trial is a 27-dimensional code signalling where the three objects are relative to its own position. As such, as the agent moves around the room it has to keep track of the three objects. This task was harder to train so we moved from the mean squared error to the cross-entropy loss and found it worked well, apart from that all details of loss and training are the same.

Between trials we randomise the position of the objects. Some portion (0.8) of the time these positions are drawn randomly (including objects landing in the same position). The rest of the time the objects were positioned so that the first object was one step north-east of the second, which itself was one step north-east of the third. This introduced correlations between the positions of the objects, while preserving their range independence - all objects could occur in all combinations.

We measured the linear NCMI between the neural activity and the 27-dimensional output code and found that each neuron was informative about a single source, as expected, it had modularised Figure 5C. However, pretend we did not know the third object existed. Instead we would calculate the linear NCMI between each neuron and those objects we know to exist. We would still find modular codes for these objects, but we would also find that the neurons that in reality code for third object, due to the correlations between object placements, are actually informative about the first and second object, so they look mixed selective! The position of the objects is always informative about each other, but by conditioning on each object we are able to remove this effect with our metric and uncover the latent modularity. But without knowing which latents to condition on we cannot proceed, and instead get lost in correlations.

