# OpenReview forum: "Range, not Independence, Drives Modularity in Biologically Inspired Representations"
_ICLR.cc/2025/Conference — ICLR 2025 Poster_

### Official Review · Reviewer_Gn2p · 2024-11-03

**Soundness:** 3
**Presentation:** 3
**Contribution:** 3
**Rating:** 6
**Confidence:** 3

**Summary:**

This paper seeks to explain why and when a population of biological or artificial neurons sometimes modularise and sometimes entangle the representation of source variables. This is a fundamental question that is highly relevant to both neuroscience and AI. The authors propose and prove a new theory emphasising the importance of the shape of empirical data distribution in extreme regions in dictating whether neurons are mixed selective or modular. Specifically if the sources to be represented are supported in all extreme regions, the neurons modularise. The application of this theory outside of linear autoencoders is tested in feed forward and recurrent neural networks, including experiments that provide explanations for discrepancies in previous neuroscience literature.

**Strengths:**

This work is original, of high quality and undoubtedly contributes to the community’s understanding of neural modularisation. The nonlinear verification of theory and additional application to neuroscience results are significant for the field and a strength of the paper. The submission is well written and clear throughout, although its clarity suffers somewhat due to the amount this submission seeks to cover.

**Weaknesses:**

- In my opinion this submission contains too much, and would benefit from more focus and time spent on fewer experiments. The appendix is already large but some experiments could be moved there.
- Figure text and panels are too small throughout.
- It is not clear how relevant encoding of the extreme points of source distributions are for computation / cognition. I.e. neurons do not just autoencode.
- The bio description of energy minimisation assumes l2 penalty is an appropriate penalisation function for modelling biology. This is a fair starting assumption, but no argument is presented about how this maps to the costs biological neurons will be seeking to minimise.

**Questions:**

- Biologically inspired linear rnns are repeatedly mentioned. Linear rnns are less like biological circuits (which are nonlinear).  Is the biologically inspired term referring to the energy costs?
150: satisfied for all w is a little unclear. The proof could
- Autoencoding seems limited as an objective for theory, in that brains perform computations over inputs and states and produce behaviour. Is it not the relevance of representing the extreme points for behaviour that is important?
- Section 2.2 assumes positive weights?
- 120 "be better" is imprecise
- Fig 1d, neuron’s angle isn’t described?
- I don’t understand the what where regression justification.
- How relevant are these results for more typical ANN experiments?  E.g classic image benchmarks or language modelling tasks?

---

> ### Author Response · Authors · 2024-11-21
> **Author Response**
>
> We thank the reviewer for their attention and reading of our paper. Below we try to address the concerns they raised.
>
> *Weakness 1: Contains too much!*
>
> *Weakness 2: Figure and text panels are small throughout*
>
> We agree that the paper is full and apologise for any lack of clarity that results. To address this, where possible, we have increased the size of figure labels and figures throughout the paper. Further, we have moved the PFC results to the appendix, making the focus of the neuroscience results more in line with the main thrust of the paper, and more coherently centred on the entorhinal cortex. With this extra space we have significantly expanded various parts of the paper, in particular the section discussing the main theorem and providing intuition and interpretation. We hope these changes have somewhat improved the clarity.
>
> *Weakness 3: How relevant are extreme points of distributions for actual neuroscience?*
>
> The reviewer is right. Our results depend on the extreme points in worrying ways. The extreme points are determined by outliers, and it seems strange for very unlikely outlying points to govern the behaviour of the optimal representations.
>
> The reason this happens in our theory is that we require the representation to perfectly decode the labels for all datapoints. If, instead, the reconstruction loss is included as a term to be minimised alongside the activity and weight energy then this failure mode would be removed. Instead of being forced to encode all outlying points, the representation could choose to ignore an unlikely extreme point, paying a small reconstruction cost for a saving in energy.
>
> We would like to theoretically understand this effect, however each of these theoretical developments takes time and we have not yet been able to develop this one. We are now working on empirical results to show this effect that we hope will be ready before the end of the discussion period, and if not by then, definitely by the time of the conference.
>
> That said, the idea that what matters for modularisation is that the support of variables is independent even if the distribution is not, seems a more reasonable conclusion than alternatives for actual neuroscience. For example, testing whether two variables are statistically independent is difficult, but checking whether they are extreme-point independent just requires tracking whether you’ve ever seen the four corners of the distribution occur. Indeed, with range-independence we are able to explain neural data that no other theory can (figure 4). We are in the process performing neural experiments in rodents to test these conclusions, though of course this is beyond the scope of the present work.
>
> *Continued - neurons do not autoencode:*
> *Question 3: Autoencoding is a limited objective, what’s up with it?*
>
> This is certainly true. Neurons are doing far more than autoencoding. However, our goal here is not a particularly accurate model of any neural circuit performing a computation. Rather, we seek a simple model of mixed selectivity vs. modularity, and why neurons might choose one or the other. A minimal model has to force the neurons to encode the variables somehow. This could be done by making the neurons perform computations with those variables, or by producing a particular behaviour as the reviewer suggests. But then theoretical analysis will likely be hard and dependent on the simplified task that was studied. Instead we try to make minimal assumptions, and just study neurons forced to encode variables. We chose autoencoding because it doesn’t make any assumptions about what the variables are being used for, just that they are represented and decodable (linearly for theoretical analysis, empirically we drop this). That this simple setup is sufficient to produce interesting behaviour that matches neural responses further justifies this choice in our mind. This of course does not negate the fact that ideally we would study theories of neurons doing meaningful computations, and this is a target for our future work. We have included a sentence in the discussion to this effect:
>
> > Additionally, our model is a purposefully simple model, there are many biologically pertinent effects whose impact on modularity would be interesting to study, such as connection sparsity, anatomical constraints, noise, or requiring additional computational roles from the network.

---

> > ### Author Response · Authors · 2024-11-21
> > **Author Response 2**
> >
> > *Weakness 4: No mapping of L2 costs to the actual costs biological neurons are trying to minimise*
> >
> > We agree that it is not obvious that an L2 cost is the appropriate one. We are not dogmatic about this, and instead simply think that neural firing is costly. As stated in the main response, to show that our conclusions generalise to other ways in which you might choose to penalise the activity we include a new appendix which shows our main result (that support independence leads to optimal modular representations) generalises to other choices of norm. We could try and get more detailed about the exact mapping of spiking onto the metabolic currency (ATP), and indeed others, such as David Attwell, have invested significant effort in calculating these energy uses. However, these quickly become complex mathematically, we prefer for the moment to take a slightly more abstracted approach that we think is broad and nonspecific enough to apply independent of the precise details.
> >
> > *Question 1: What’s going on with biological linear RNNs?*
> >
> > The biological indeed refers to the energy regularisation. Real biological RNNs are of course non-linear. Rather, our theory applies to linear RNNs with energy constraints, and we derive some interesting results regarding modularisation in these systems. Apologies for the confusion, we have tried throughout to instead refer to them as ‘linear RNNs with biological constraints’
> >
> > *Question 4: Section 2.2 assumes positive weights?*
> >
> > Apologies, yes, parts of the intuitive argument in section 2.2 rely on using positive weights and we now include that. Thank you for catching that.
> >
> > *Question 5: ‘be better’ is imprecise.*
> >
> > Agreed, corrected to ‘uses less energy’.
> >
> > *Question 6: Fig 1d neuron’s angle not described?*
> >
> > Apologies, we’ve included a longer description in the caption to define this concept:
> >
> > > Here the y axis measures how mixed the representation is by quoting the largest angle between a neuron’s weight vector and one of the source axes.
> >
> > *Question 7: I don’t understand the what-where justification*
> >
> > Apologies for the confusion. The what-where regression tries to test the qualitative trends outlined at the end of section 2 for when nonlinear representations of variables (the two variables here are what and where) should modularise. We therefore test three datasets. The first removes a corner from the joint range of what and where (i.e. position 1 (where) and shape 1 (what) don’t co-occur, so there is a point missing from the co-range of what and where). We then test how the modularity of the optimal representation varies as the size of the missing corner increases. The second test (green) changes the distribution of the data such that more data is drawn from the diagonal (e.g. what = 3 and where = 3 is more common than what = 2 and where = 3), finally the last test removes data from the diagonal (what = 5, where = 5) rather than the corners. We find qualitative trends that match the theoretical predictions. Removing data from the corners is more effective than both correlations that don’t change the range and removing diagonally positioned data for producing mixing for the same amount of induced source multiinformation.
> >
> > *Question 8: How relevant are these results for normal ANN stuff (language or images)?*
> >
> > We would likely proceed much as we did in the last panel of figure 2, which is an application of our theory to a complex image dataset. Apologies if is was not clear this was a real image dataset, we now include text to highlight this.
> >
> > > We study the performance of QLAE trained to autoencode a subset of the Isaac3D dataset (Nie 2019), a naturalistic image dataset with well defined underlying latent dimensions.
> >
> > As such, we would assume that the measured representation is encoding a particular set of latent variables, then reason about how the theory suggests the distribution of those sources latents might lead them to be optimally encoded in a modular or mixed fashion. This has many obvious weaknesses (what are the latents? What if it is not just encoding but is computing with the latents, won’t that affect things?), but presents a good start that is already predictive, as in figure 2.
> >
> > We would like to thank the reviewer for their attention towards our work.

---

> ### Author Response · Authors · 2024-11-25
> **Reminder**
>
> Thank you again for your valuable feedback. As the end of the discussion period is approaching, we are hoping to hear your thoughts on our response.
> We hope that we have addressed your comments, and we would greatly appreciate it if you were willing to consider increasing your score if you are satisfied with our response.

---

> > ### Comment · Reviewer_Gn2p · 2024-11-25
> >
> > I thank the authors for their detailed and helpful responses, they have clarified a number of my concerns. I think the changes to the paper strengthen it, making its contributions both clearer and more impactful to the field. As such I believe this paper should be accepted and I have increased my score accordingly.
> >
> > Reasons for not increasing the score further:
> > - I still have some concerns regarding the general applicability and importance of the extreme points outside of a perfect linear reconstruction constraint. I look forward to the authors' upcoming work on this.
> > - I am unable to award a 7.

---

> > > ### Author Response · Authors · 2024-11-28
> > > **Final Response**
> > >
> > > Thank you very much for your feedback!
> > >
> > > We will include the remaining parts of our responses to you in clarifying details in the paper for its final version.
> > >
> > > We have unfortunately not been able to produce the reconstruction error simulations in time for the rebuttal period, so those will have to be included in a later version, both if it is accepted and if not. A final thought is that our image experiments in figure 2 already have a reconstruction loss element, and demonstrate that the main ideas seem to generalise to that setting.
> > >
> > > Thank you again for your engagement with our work!

---

### Official Review · Reviewer_1fwm · 2024-11-04

**Soundness:** 3
**Presentation:** 2
**Contribution:** 3
**Rating:** 6
**Confidence:** 4

**Summary:**

This paper studies the conditions under which modularity should arise in optimal representations. The authors developed a mathematical theory of when a modular representation should be favored in a linear autoencoder. They found that modularity should appear when the support of the sources is “sufficiently spread”. The paper also presented simulation results to show that some of theoretical results may be generalizable to non-linear problems. The later sections of the paper applied these theoretical ideas to explain some experimental observations from neurophysiological experiments in cognitive tasks.
The paper makes several interesting points about when modularity should be favored, and applies the theory to several examples.

**Strengths:**

Originality: The theoretical part of the work builds up prior work by Whittington et al, 2023 and other studies. Previous work by Whittington et al, 2023 assumed mutual independence of the sources. In the current work, the authors show that, with several additional assumptions, “sufficient spread” of the factors of variation can also lead to modular representation. The theory has some new elements, although it is a bit incremental. The application to the several neuroscience problems seems to be new.

Quality: The paper considered both linear and nonlinear cases. This is a strength. For the former, analytical results were provided. For the latter, some preliminary numerical results were given.
The paper also considered several neuroscience applications. This may also be seen as a strength.


Clarity: The overall structure of the paper is clear. Some intuitions behind the theory were provided.


Significance: The question of when modularity arises in optimal representation is an interesting one and we still lack a clear understanding. This work made a few interesting points on this problem.

**Weaknesses:**

The writing needs improvements throughout the paper. In particular, the description of the theory can be substantially improved. For example, Theorem 2.1 should be made more accessible.


While several applications are attempted, each application appears to be preliminary. If the model predictions and experimental tests can be made more rigorous, that would strengthen the paper.

In Section 5, there are some qualitative differences between the model predictions and the data. As the paper pointed out, Panichello and Buschman (2021) showed that a substantial fraction of the neurons were tuned to both colors, contradicting a key prediction of the model. This seems to be a more important feature of the data compared to the issue of orthogonality v.s. non-orthogonality.

Looking at the math, the theory appears to only work for scalar variables. Can it be applied to circular variables? If the answer is no, the applications to real data would be questionable. In section 5, color is sampled from a color-wheel.

Based on the way the theory is written, the results seem to rely the assumption that the energy (or cost) is a quadratic function of neural activity. If the cost scales linearly with neural activity, would the theoretical results change fundamentally? Assuming a linear scaling could make sense biologically, as the metabolic cost may scale linearly with the number of spikes.

Relevant earlier theoretical literature on grid cell modularity was not cited/discussed (e.g., by Fiete/Burak et al, and Wei/Prentice/Balasubramanian).


It is difficult to understand what is really going on in Fig. 1. Fig.1 may come out too early and the results need to be better unpacked.


The theory seems to ignore biologically-relevant noise.

**Questions:**

What are the assumptions about the noise in the system being studied? How does noise affect the theoretical results?

In section 5, the authors seems to be equating orthogonality with modularity. Am I understanding this correctly? If this is the case, can the authors unpack the idea?

Can they authors unpack the results in Fig. 5c? Was that an actual simulation or just a schematic?

The definition of the matrix in Eq. 10 is unclear. Please clarify.


Can the authors explain what the first part of the title mean?

In the abstract, it is stated that “From this theory, we extract and validate predictions…” What does “extract” predictions mean? [minor point]

---

> ### Author Response · Authors · 2024-11-21
> **Author Response**
>
> We thank the reviewer for their careful and detailed review of our work that helped us significantly improve it. Below we hope to address some of their concerns.
>
> *Weakness 1: Writing could be improved, including theorem introduction in theorem 2.1*
>
> We’re sorry for the unclear writing.To remedy this we have significantly changed the buildup to theorem 2.1 by adding additional details to section 2.2 clarifying the intuition that can be used for why range independence leads to modularity, which now reads:
>
> > The key takeaway lies in how $b_j$ is determined by a joint minimization over $s_1$ and $s_2$ (5). Assuming positive $w_{j1}$ and $w_{j2}$; then if $s_1$ and $s_2$ take their minima simultaneously, as in the middle row of Fig1a, then mixed bias must be large:
> \begin{equation}
> b_j = -\min_{s_1, s_2} \left[ w_{j1} s_1 + w_{j2} s_2 \right] = -  \min_{s_1}[w_{j1} s_1] - \min_{s_2} [w_{j2} s_2] = b_{j'} + b_{j''}
> \end{equation}
> And the energy of the mixed solution will always be worse than the modular, since $b_j^2 = (b_{j'} + b_{j''})^2 > b_{j'}^2 + b_{j''}^2$. Alternatively, mixing will be preferred when $s_1$ and $s_2$ do not take on their most negative values at the same time, as in the bottom row of Fig 1a, since then $b_j$ does not have to be as large to maintain positivity, and the corresponding energy saving satisfies the key inequality (6).
>
> Further, we have added the following clarifying sentences after the introduction of theorem 2.1 to link it to the previously discussed intuition.
>
> > These inequalities come from the difference in activity energy between a modular and mixed solutions, just like the intuition we built up in Section 2.2, and in particular Equation 6.
>
> More broadly, we have moved one of the neural analysis sections to the appendix and used the space to expand the other sections. We have expanded the discussion of the effects of range independence (new section 2.4), grown the figure labels to make it clearer, and rewritten the mixed selectivity part of the discussion. Our most pertinent changes were to figure 1, in answer to another of your comments:
>
> *Weakness 7: Unclear what’s going on in figure 1, came out too early*
>
> Apologies for Figure 1. We have now completely re-worked it, moved it much later in the paper, and added a much extended caption to try and more clearly explain it. The new intuitive figure (new fig 1A) focuses on the difference between a mixed encoding of a range-independent and range-dependent pair of variables, following more closely the discussion in the intuitive section. Further, we have made it larger to make it clearer, and added to the caption to walk the reader through our argument in the sketch figure (1a), and the definitions of all quantities in the data figures (1b and 1c). We thank you for pointing to these shortcomings.
>
> *Weakness 2: Applications to data appear to be preliminary*
>
> *Weakness 3: Non-neuron aligned subspaces in PFC, what’s going on there?*
>
> *Relatedly: Question 2: What is the relationship between modularity and orthogonality?*
>
> We agree with the reviewer, in the original submission the PFC section was slightly too preliminary, and we merged orthogonality and modularity too quickly. As such, we have moved these results to the appendix and replaced it with a section that focuses on how our theory impacts interpretations of the role of mixed selectivity from data, which we would be grateful if the reviewer considered. We thank the reviewer for prompting us to think more carefully about this, and we are currently running further analysis to address their and other concerns.
>
> However, regarding the entorhinal mixed selectivity results, we disagree with the reviewer and consider them thorough. We hope that our response to your next question will convince you of the thoroughness of the results.

---

> > ### Author Response · Authors · 2024-11-21
> > **Author Response Part 2**
> >
> > *Q3: Can the authors unpack the results in Fig. 5c? Was that an actual simulation or just a schematic?*
> >
> > Yes! Figure 5c (figure 4c in new submission) is a real simulation result! We trained an RNN model to output an agent’s current position and its displacement from reward as an agent moved around a 2D grid world. We found that, in line with the theory, when reward and position are range independent (i.e. when knowing the value of one variable does not constrain the allowed set of values for the other variable, this is true if reward is at least partially randomly scattered, since then reward could occur in all positions)  their representation was perfectly modular, whereas when their ranges are not independent (some positions are never rewarded) the representations mix. This matches all known mixing and modularity results from entorhinal cortex - results that could not be explained before our theory. As such we consider these results complete and a ringing endorsement of the theory. In fact, we consider the theoretical side of the work so complete that we are moving on to testing them experimentally, in new neural recordings (though this is of course beyond the scope of this paper). Apologies for this not being clear.
> >
> > *Weakness 4: Theory only works for scalar variables, but PFC example is 2D at least, what’s up with that?*
> >
> > *Weakness 5: Seems to rely on L2 scaling of energy. If scaling is linear does stuff change massively?*
> >
> > We thank the reviewer for pointing our attention to these shortcomings. In our new submission we include two new proofs that show that our core results generalise to cover both of these cases. First, in Appendix C, we show that support independent variables are optimally modular if you use any Lp norm as the activity loss for p > 1, and if p = 1 then the modular solution is optimal, but there might be other non-modular but orthogonal solutions that are equally good. Further, in Appendix D, we show that the same support independent results apply to multidimensional variables, i.e. a set of support independent multidimensional variables should also be modularised, again for all choices of Lp activity norm. We hope that this convinces you of the generality of our theory.
> >
> > Regarding the appropriate choice of activity norm we acknowledge that it is unclear. L1 expresses the idea, as the reviewer says, that counting spikes might be the reasonable cost. L2 instead, expresses the intuitive idea that an increase of spiking of 10Hz when you’re already spiking at 100 Hz might be more costly than the same increase from 0Hz. We’re not too dogmatic about which is the best choice, so we are thankful that our main point, that support properties are the drivers of modularity, extend to all such reasonable choices of norm.
> >
> > *Weakness 6: Doesn’t cite relevant work on grid cell modularity*
> >
> > We thank the reviewer for their suggestion. Both the papers you mentioned hard code grid cell modularity, and so do not examine what the necessary conditions—either in the data distribution or in biological constraints—are to observe modularity or not. These papers however address different questions: Burak & Fiete provides a computational model of how multi-modular grid cells might path-integrate (e), whereas (Wei/Prentice/Balasubramanian and similar work by Stemmler) shows how multi-modular codes are particularly  good for spatial coding and discuss how various aspects of the code that might be optimised to store spatial information. For our purposes, it is important to note that neither of these works (or others) normatively explains why grid cells form modules, over any other coding scheme, they just show that such a scheme is good. In our work, instead we show why grid modules align with the neuron basis, something assumed in all other works.
> >
> > We have added this discussion to our relevant work section:
> >
> > > One of the many surprising features of grid cells is their modular structure: each grid cell is a member of one module, of which there are a small number in rats. Grid cells from the same module have receptive fields that are translated versions of one another (Stensola et al., 2012). Previous work has built circuit models of such modules showing how they might path-integrate (Burak and Fiete, 2009), or has assumed the existence of multiple modules, then shown that they form a good code for space, and that parameter choices such as the module lengthscale ratio can be extracted from optimal coding arguments (Mathis et al., 2012; Wei et al., 2015). However, neither of these directions shows why, of all the ways to code space, a multi-modular structure is best. Dorrell et al. (2023) study a normative problem that explains the emergence of multiple modules grid cells as the best of all possible codes, but their arguments for the emergence of multiple grid modules are relatively intuitive. In this work we are able to formalise parts of their argument, linking it cohesively to theories of modular and mixed coding.

---

> > > ### Author Response · Authors · 2024-11-21
> > > **Author Response Part 3**
> > >
> > > *Weakness 8: Ignores biologically relevant noise*
> > >
> > > *Relatedly - Question 1: What are the assumptions on noise?*
> > >
> > > Our model is indeed a very simple model, and purposefully so! It allows us to understand it in detail and extract surprising predictions that match neural data. Biological noise is likely an important concern. In our work we do not consider it, rather we consider the optimal noise-less tuning curve. Most work that uses artificial neural networks to predict neural activity makes a similar assumption, and it is interesting how well this appears to be working. Nevertheless, we think that noise is an important direction and have added the following to our limitations section to discuss this and other effects:
> > >
> > > > Additionally, our model is a purposefully simple model, there are many biologically pertinent effects whose impact on modularity would be interesting to study, such as connection sparsity, anatomical constraints, noise, or requiring additional computational roles from the network.
> > >
> > > *Question 4: What is going on with the matrix S?*
> > >
> > > Using the matrix notation just summarises the system of equations from earlier proof, and helps build the other interpretation of the proof - when the ellipse (defined by S) is fully contained within the convex hull of the data or not. We have tried to rephrase the maths in that section to make it clear where that matrix comes from. In particular, we have added an extra row and column to the definition so the completion is clearer, moved its definition closer to its natural introduction, and highlighted where it pops up naturally. In short, the matrix is minus the covariance matrix of the sources with its diagonal removed and replaced by the square of the min of each of the sources. It defines the ellipse that governs the modularising behaviour.
> > >
> > > *Question 5: What’s up with corner cutting in the title*
> > >
> > > The corner cutting was meant to be about how modularity can be turned off/on depending on whether the corner of the data distribution is cut off or not. We agree that this is very unclear! So we have changed the title to focus on the main point of the paper that generalises to other norms and multidimensional variables:
> > >
> > > > Range not statistical independence drives modularity in biological representations
> > >
> > > (despite the fact this is only a sufficient result, unlike the main result, which is also necessary)
> > >
> > > *Question 6: What’s up with ‘extracting conclusions’*
> > >
> > > Sorry for the confusing wording. We simply meant that we looked for qualitative trends in the theory that we could test in the nonlinear settings. We have changed the wording to clarify this.
> > >
> > > Finally, we would like to emphasise the development of our work from that of Whittington et al. ‘23. We assume no additional assumptions relative to that work, yet are able to derive much weaker conditions on the sources that lead to modularisation. It is only thanks to this weakening of conditions that we are able to understand modularisation in RNNs (fig 3), neural data from entorhinal cortex (now fig 4) and make subtle conclusions about potential sources of measured mixed selectivity (new fig 5). Further, our conditions are necessary and sufficient, and able to predict the degree of modularisation or mixing much more accurately than the work of Whittington et al. ‘23.
> > >
> > > Thank you again for your time and attention!

---

> > > > ### Comment · Reviewer_1fwm · 2024-12-03
> > > >
> > > > Thanks for detailed response. The revision addressed some of my concerns. It helps to analyze fewer examples in greater depth. I still have substantial reservations about the results in Section 5 and the restrictive assumptions behind the theory. But given the improvement of the paper, I am happy to raise my score to 6.

---

> ### Author Response · Authors · 2024-11-25
> **Reminder**
>
> Thank you again for your valuable feedback. As the end of the discussion period is approaching, we are hoping to hear your thoughts on our response.
> We hope that we have addressed your comments, and we would greatly appreciate it if you were willing to consider increasing your score if you are satisfied with our response.

---

### Official Review · Reviewer_a7qJ · 2024-11-05

**Soundness:** 3
**Presentation:** 4
**Contribution:** 3
**Rating:** 6
**Confidence:** 3

**Summary:**

This paper investigates why neural representations in biologically inspired networks sometimes form modular structures, where each neuron encodes a single variable, and other times create mixed-selective representations. The authors develop a theory that predicts when modularization will occur in networks optimized for energy efficiency with nonnegative firing rates. They derive necessary and sufficient conditions for modularity based on the spread of source variables. The theory is validated in both linear and nonlinear networks. The theory provides a cohesive explanation for the conflicting findings in the prefrontal cortex and entorhinal cortex data from neuroscience studies.

**Strengths:**

1. The paper introduces a novel theory that precisely predicts necessary and sufficient conditions for modular representations in biologically inspired networks, extending previous work beyond statistical dependencies.

2. The mathematical formulation is rigorously derived, and validated across various neural network architectures and experiments.

3. The theory provides explanations for conflicting neuroscience findings and has close links to biologically plausible architectures and brain representations.

4. The paper provides a cohesive theory for understanding modularity in neural representations, with implications for both interpreting biological neural data and guiding the design of artificial neural networks for better interpretability and efficiency.

5. The paper is well-written, presenting complex theoretical concepts with clarity and intuition.

**Weaknesses:**

1. The experiments use nonnegative activities in neural networks, which aligns with biological plausibility, but it would be valuable to discuss inhibitory neurons in the brain and how inhibition might relate to the theory and findings.

2. While the L2 norm of firing rates and weights is a reasonable approximation for biological energy, other biological constraints (e.g., sparse connectivity, synaptic range, anatomical structure, and decoding flexibility) may also play a role.

3. It’s unclear how the theory would extend to more complex datasets. For example, what would the different conditions/variations in source variables mean in naturalistic data (e.g., natural images, audio, text)? How might we approximate "spread", and quantify modularity conditions in such stimuli?

4. The discrepancies in prefrontal working memory modeling and the brain data could have further explanations, particularly why some neurons tune to both colors despite orthogonal encoding and why exact subspace angles were not obtained.

**Questions:**

1. Beyond prefrontal working memory and the entorhinal cortex, does the theory generalize to other modular representations in the brain?

2. If decoding flexibility were considered as a biological constraint, how might it impact the theory and its predictions?

---

> ### Author Response · Authors · 2024-11-21
> **Author Repsonse**
>
> We would like to thank the reviewer for their careful, and largely positive, review of our work. Below we hope to address some of their concerns.
>
> *Weakness 1: Non-negative activities, what about inhibitory neurons?*
>
> The reviewer is right that in our simple model we do not split between excitatory and inhibitory neurons. We do this as we are interested in the simplest possible account of modularity and so only use the fewest constraints. We see this as a real positive.  Thus our theory is largely modelling how excitatory neurons (cortical pyramidal neurons) should encode variables as simply as possible. With that in mind, it is interesting that by focusing on excitatory neurons we can already get non-trivial results that match neural data. Nevertheless, we agree with you that incorporating inhibitory neurons may lead to further non-trivial situations in which modularity occurs - we leave this for future work.
>
> *Weakness 2: L2 is cool, but why no other biological constraints (sparse connectivity, synaptic range…)?*
>
> This is a good question. Ultimately we are looking for the simplest set of biological constraints that lead to modularity. You’re right that to have the fullest picture of how biological constraints interact with representation, then including things like sparse connectivity and synaptic range would be great. But again, we’re looking for the simplest set of constraints that induce modularity. It is surprising that simple energy minimisation in a linear autoencoder got us this far, yet was quite hard for us to understand fully. Further understanding things like sparse connectivity and synaptic range is in our overall mission, but that is a big ambition and will be other papers. We now include a statement in the discussion to this effect:
>
> > Additionally, our model is a purposefully simple model, there are many biologically pertinent effects whose impact on modularity would be interesting to study, such as connection sparsity, anatomical constraints, noise, or requiring additional computational roles from the network.
>
> With particular reference to the use of L2 activity loss (rather than other reasonable choices), in our new submission we include an additional theoretical result that shows that range-independence is sufficient to drive modularity for a wide range of choices of activity norm (new appendix C).
>
> *Weakness 3: How could the theory extend to more complex datasets, e.g. images and text*
>
> We would likely proceed much as we did in the last panel of figure 2, which is an application of our theory to a complex image dataset. Apologies if this was not clear, we now highlight this in the main text:
>
> > We study the performance of QLAE trained to autoencode a subset of the Isaac3D dataset (Nie 2019), a naturalistic image dataset with well defined underlying latent dimensions.
>
> So to proceed we would assume that the measured representation is encoding a particular set of latent variables, then reason about how the theory suggests the distribution of those sources latents might lead them to be optimally encoded in a modular or mixed fashion. This has many obvious weaknesses (what are the latents? What if it is not just encoding but is computing with the latents, won’t that affect things?), but presents a good start that is already predictive, as in figure 2.
>
> *Weakness 4: Why do the PFC results not match? Both in subspace angle and neuron alignment*
>
> Yes, this is confusing. We have a lot of ideas for exploring this discrepancy (in brief: the numerical subspace alignment might be fixed by changing hyperparameters of the task; and we are diving into the original neural data to test some hypothesis about what might cause the mixing on a neural level). These directions require more work and, and so we have decided to move this section to the appendix and replace it with a discussion of other potential causes of mixed selectivity, beyond the currently predominant theories that our work suggests. Thank you for raising these issues.
>
> *Question 1: Does the theory generalise to other brain areas?*
>
> Despite moving the PFC section to the appendix, we remain hopeful that our work applies broadly to neural coding. We are interested in testing it in sensory and motor areas, as well as refining the removed PFC results. It is known that cells in parietal cortex, for example, that neurons are highly responsive for single tasks (Lee, Krumin, Harris, Carandini, 2022) and not others (they are modular), while in the prefrontal cortex there are modular neurons for things like the abstract structure of a sequence (Shima & Tanji 2007). However, in this study, and most other neuroscience studies, they do not have diversity in their task structure to test for concepts like range independence. We are actively collaborating with experimentalists to run studies which directly test our theoretical claims in a variety of brain regions, and we hope to be able to answer this question in the coming years.

---

> > ### Author Response · Authors · 2024-11-21
> > **Author Response Part 2**
> >
> > *Question 2: What would happen if decoding flexibility were considered as another constraint?*
> >
> > Unfortunately, we’re not sure what the reviewer means by ‘decoding flexibility’, so feel badly placed to answer this question. The setting we consider is already somewhat flexible in that the readout can decode one variable independently of the behaviour of all others, so in that sense it is maximally flexible. Another definition of a ‘decoding flexibility’ might be more flexible decoders, i.e. nonlinear ones. That would imply a different set of constraints that lead to modularity, but is of the flavour that we tried to test with our nonlinear networks - there the decoders are arbitrary nonlinear networks, and yet we still find that our theory is predictive. Finally, we thought flexible decoders might mean flexibly decoding functions of the sources rather than just the sources themselves. In this case, it would depend on the precise implementation, but alone this does not seem sufficient to cause modularity: as long as the variables are linearly decoded you can decode all linear functions of the two variables, and a sufficiently flexible readout could predict all functions of the variables. Before commenting further it would be useful to understand more the type of flexibility the reviewer was interested in.
> >
> > We thank the reviewer again for their time and attention.

---

> ### Author Response · Authors · 2024-11-25
> **Reminder**
>
> Thank you again for your valuable feedback. As the end of the discussion period is approaching, we are hoping to hear your thoughts on our response.
> We hope that we have addressed your comments, and we would greatly appreciate it if you were willing to consider increasing your score if you are satisfied with our response.

---

> > ### Comment · Reviewer_a7qJ · 2024-11-27
> >
> > Thank you for the detailed and thoughtful responses. They have addressed most of my concerns, particularly regarding the scope of the theory, the inclusion of additional theoretical results, and clarifications on the naturalistic image dataset. Incorporating these responses into the updated manuscript will strengthen the paper.
> >
> > However, I still believe that the paper would benefit from more empirical neuroscience evidence, such as aligning the model more closely with PFC representations and including additional neural data to support the theory.
> >
> > I would increase my score to 7 if allowed.

---

> > > ### Author Response · Authors · 2024-11-28
> > > **Final Response**
> > >
> > > Thank you for your response!
> > >
> > > We agree, more neuroscience data would help, and we're currently actively looking at the puzzling PFC single neuron responses.
> > >
> > > We are pleased to say, however, that one of the discrepancies between our theory and the data has now been removed. Previously the numerical alignment between subspaces was different in data vs. theory. As we suggested, we played with the hyperparameters (changed the relative weighting of weight vs. activity loss from 1 to 0.3, and added a delay period of length 2 rather than 1) and the two now align.
> > >
> > > Thank you again for your engagement with our work!

---

### Author Response · Authors · 2024-11-21
**General Comment**

We’d like to thank all the reviewers for their time and attention to our work, and for their enthusiasm. Their comments have helped us to improve and sharpen the paper.

In general the reviewers praised the rigorous analytical results (a7qJ, 1fwm), empirical tests in nonlinear networks (a7qJ, Gn2p), links to neuroscience (a7qJ, Gn2p), clarity of presentation (a7qJ, Gn2p), and choice of question (1wfm). In addition, the reviewers shared concerns that we have tried to address. To highlight the largest changes in the new submission, we have:
1. Moved the section on PFC to the appendix
2. Extended the entorhinal section to comment on potential sources of mixed-selectivity
3. Significantly clarified the main theorem presentation.
4. Added more theorems that show our core results generalise to other choices of activity norms and multidimensional latents.

We now run through the shared concerns in more detail.

**PFC DATA** First, there was concern that, of the two comparisons to neural data, the PFC comparison was weaker. The original submission contained both a numerical mismatch of the subspace alignment between the theoretical and empirical representations (a1qJ), and the neural level modularisation results do not at first blush agree with our theory (1fwm, a1qJ). Further, we previously elided concepts of orthogonality and modularity in confusing ways (1fwm). While we have lots of thoughts for how to improve and develop these results, we largely agree with the reviewers on these points and have concluded that, at this stage, the results are slightly too preliminary.

As a result we have moved the section on the PFC to the appendix, and have replaced it with a new section that expands how our theory contributes to the ongoing debate over the role of mixed selectivity in the brain. This highlights how our theory leads to multiple additional interpretations of mixed selectivity of neurons, beyond the prevalent view that mixed selectivity exists to enable flexible downstream categorization (Rigotti et al. 2013).

**Overly Simplistic Model** Second, many reviewers pointed to the simplicity of our model, asking about including inhibitory neurons (a1qJ), other biological constraints (sparse connectivity, synaptic range, anatomical structure)) (a1qJ), biological noise (1fwm), and our focus on an autoencoding objective (Gn2p). These are all reasonable details to include. However, in contrast, we see the simplicity of our setting as a good thing! We are interested in understanding why neurons sometimes choose to mix their representations, and other times not. To do this we build the simplest model that can capture the observed phenomena. In order to make the representation encode the variables we use autoencoding, which implies minimal additional structure. Then we find that just energy minimsation leads us to rich phenomenology capable of explaining neural data in ways that were not previously possible, making testable experimental predictions. Each of these other effects would definitely be interesting to study, and other works have used such effects to derive interesting modularity conclusions (such as Liu, Khona, Fiete, & Tegmark. 2023), but with just these simplest set of constraints we can go surprisingly far!

**L2 Activity Norm** Some particular concerns reviewers pointed to were the use of the L2 activity norm rather than some other norm (1fwm, Gn2p), and the limitation to scalar variables (1fwm). In our updated version we present new results that show that our core result (that support independence drives modularisation) generalises to other choices of activity norm (Appendix C in new submission) and to multidimensional variables (Appendix D). In particular, this means that the particular choice of activity norm is not vital, as long as the simple idea that spiking is energetically costly is used.

**More Complex Tasks** Finally, reviewers wondered whether our work generalised to image or language tasks (a7qJ, Gn2p). First, we point out that figure 2 included experiments performed on a standard image dataset, Isaac 3D, and we now highlight this in the paper. This gives a model for how our work might be applied to these settings. I.e. if you think a particular representation contains information about a set of variables, for example the latent factors in Isaac 3D, then our theory gives tools to think about what properties of those variables will determine whether they are represented in a modular or mixed fashion. We are actively trying to use these results to think about modularisation in more complex tasks, such as in LLMs, by making guesses about the encoded latent variables, and seeing whether we can predict the modularity of their representation from the support properties. This, however, will be for future work, as it involves significant extra work.

Overall, we would like to thank the reviewers for their time and effort, and look forward to hearing what they have to say in response.

---

> ### Author Response · Authors · 2024-11-28
> **Final PDF Change**
>
> To briefly explain our final submission edit, we fixed a few typos, and changed some task parameters in order to make the PFC ersult numerically align with the data.
>
> How aligned the two subspaces are in our PFC model networks depends on critical hyperparamters in the theory such as the weighting of weight vs. activity loss, and the length of the delay period. We played a little with these and found a set that agreed numerically with the data. The crucial prediction remains the same however: if the stimuli are range-independent then, regardless of these choices of hyperparameters, the subspaces are orthogonal. If they are range-dependent then the subspaces can align, and how much they do depends on hyperparameter choices in our theory.

---

### Meta-Review · Area_Chair_nLV9 · 2024-12-09

**Metareview:**

This work tackles a timely subject in distributed computing in the brain: modularity. The work aims to provide a theoretical framework that can identify when and why modularity occurs. The theory itself builds off of the analysis of optimal encoding of a set of inputs, with the general constraints that the encoding must be "simplest" by minimizing the norms of the weights, and that the encoding variables are non-negative. Both assumptions are used to identify the key result, formulate the result in a way where comparing a convex hull of the data to a theory-driven ellipse results in an understanding of when inputs should be mixed or not. The paper itself is very dense with a significant appendix detailing much of the theoretical ideas and derivations. The primary result of the work was that the concept of data spread could be considered as important as statistical independence between variables that are encoded.

The reviewers mentioned a number of points where clarity could be improved, and the authors in response amended the manuscript and included additional analyses for extending the model and adding more experimental results. Some of the more central points might be more important to the assessment of this manuscript. One reviewer raised the question of why en encoder framework, independent of task requirements etc. is the right framework in which to view modularity. Similarly, the question from another reviewer that the model depends too much on the linear setting. I believe these concerns actually speak to a deeper assumption: what are the chosen inputs? There are multiple levels in which tasks and performance can be modularized or not. In some ways receptive fields are "non modular" in the space of localized point sources. The entire endeavor rests on definition of what is a "single" input and "single" processing unit (here chosen to be a single neuron). These are largely assumed and I think instantiated in a number of different questions the reviewers had in different ways.

Beyond these concerns, another recurring theme was the length of the manuscript. It is true the authors have done much work, but there is also significant duplication of information. I agree that a more focused paper would likely fair much better for the shorter format here.

In all, while this is a borderline case, the reviewers in the end felt the merits outweighed the shortcomings and recommend accepting this work.

**Additional Comments On Reviewer Discussion:**

The responses were pretty cut-and-dried.

---

### Decision · Program_Chairs · 2025-01-22

Accept (Poster)